# Incentives in Private Collaborative Machine Learning

**Rachael Hwee Ling Sim**[1] , **Yehong Zhang**[2] , **Trong Nghia Hoang**[3]
**Xinyi Xu**[1] , **Bryan Kian Hsiang Low**[1] , and **Patrick Jaillet**[4]
[1] Department of Computer Science, National University of Singapore, Republic of Singapore
[2] Peng Cheng Laboratory, People's Republic of China
[3] School of Electrical Engineering and Computer Science, Washington State University, USA
[4] Dept. of Electrical Engineering and Computer Science, MIT, USA
[1]{rachaels,xuxinyi,lowkh}@comp.nus.edu.sg, [2]zhangyh02@pcl.ac.cn
[3]trongnghia.hoang@wsu.edu, [4]jaillet@mit.edu

## Abstract

Collaborative machine learning involves training models on data from multiple parties but must incentivize their participation. Existing data valuation methods fairly value and reward each party based on shared data or model parameters but neglect the privacy risks involved. To address this, we introduce *differential privacy* (DP) as an incentive. Each party can select its required DP guarantee and perturb its *sufficient statistic* (SS) accordingly. The mediator values the perturbed SS by the Bayesian surprise it elicits about the model parameters. As our valuation function enforces a *privacy-valuation trade-off*, parties are deterred from selecting excessive DP guarantees that reduce the utility of the grand coalition's model. Finally, the mediator rewards each party with different posterior samples of the model parameters. Such rewards still satisfy existing incentives like fairness but additionally preserve DP and a high similarity to the grand coalition's posterior. We empirically demonstrate the effectiveness and practicality of our approach on synthetic and real-world datasets.

## 1 Introduction

Collaborative *machine learning* (ML) seeks to build ML models of higher quality by training on more data owned by multiple parties [47, 62]. For example, a hospital can improve its prediction of disease progression by training on data collected from more and diversified patients from other hospitals [6]. Likewise, a real-estate firm can improve its prediction of demand and price by training on data from others [9]. However, parties have two main concerns that discourage data sharing and participation in collaborative ML: (a) whether they benefit from the collaboration and (b) privacy.

Concern (a) arises as each party would expect the significant cost that it incurs to collect and share data (e.g., the risk of losing its competitive edge) to be covered. Some existing works [47, 51], among other data valuation methods,[1] have recognized that parties require incentives to collaborate, such as a guaranteed *fair* higher reward from contributing more valuable data than the others, an *individually rational* higher reward from collaboration than in solitude, and a higher total reward (i.e., *group welfare*) whenever possible. Often, parties share and are rewarded with information (e.g., gradients [58] or parameters [47] of parametric ML models) computed from the shared data. However, these incentive-aware reward schemes expose parties to privacy risks.

---

[1]Data valuation methods study how much data is worth. As explained in [46], a party's data is first valued independently using a performance metric (e.g., see Def. 3.1 later) and then relative to the data contributed by others (e.g., Shapley value (Sec. 4)). The latter value is helpful to (i) model interpretability and (ii) deciding how much to compensate the data owners *fairly*.

On the other hand, some *federated learning* (FL) works [34] have addressed the privacy concern (b) and satisfied strict data protection laws (e.g., European Union's General Data Protection Regulation) by enforcing *differential privacy* (DP) [1, 36] during the collaboration. Each party injects noise before sharing information to ensure that its shared information would not significantly alter a knowledgeable collaborating party's or mediator's belief about whether a datum was input to the algorithm. Injecting more noise leads to a stronger DP guarantee. As raised in [64], adding DP can invalidate game-theoretic properties and hence affect participation. For example, in the next paragraph, we will see that adding DP may lead to the collaboration being perceived as unfair and a lower group welfare. However, to the best of our knowledge (and as discussed in Sec. 7 and Fig. 5), there are no works that address both concerns, i.e., ensure the *fairness*, *individual rationality*, and *group welfare* incentives (see Sec. 4), alongside privacy. Thus, we aim to fill in this gap and design an incentive-aware yet privacy-preserving reward scheme by addressing the following questions:

**If a party (e.g., hospital) requires a stronger DP guarantee, what should the impact be on its valuation and reward?** Our answer is that, on average, its valuation and reward should decrease. Intuitively, it is unfair when this party gets a higher valuation due to randomness in the DP noise. More importantly, parties require guaranteed higher rewards to consider a weaker privacy guarantee [22, 64] which will help maximize the utility of the collaboratively trained model(s). As observed in [14, 65], the weaker the DP guarantee, the smaller the loss in model accuracy from enforcing DP. Thus, we will (i) assign a value to each party to enforce a *privacy-valuation trade-off* and incentivize parties against unfetteredly selecting an excessively strong DP guarantee,[2] and (ii) flexibly allow each party to enforce a different DP guarantee without imposing a party's need for strong DP on others. This new perspective and its realization is our main contribution.

**To enforce a privacy-valuation trade-off, how should DP be ensured and a party's data be valued (Sec. 3)?** Initially, valuation using validation accuracy seems promising as the works of [18, 25] have empirically shown that adding noise will decrease the valuation. However, parties may be reluctant to contribute validation data due to privacy concerns and disagree on the validation set as they prioritize accurate predictions on different inputs (e.g., patient demographics). So, we revert to valuing parties based on the quality of inference of the model parameters under DP. Bayesian inference is a natural choice as it quantifies the impact of (additional DP) noise. In Sec. 2, we will explain how each party ensures DP by only sharing perturbed *sufficient statistic* (SS) with the mediator. The mediator values the perturbed SS by the *surprise* it elicits relative to the prior belief of model parameters. Intuitively, noisier perturbed SS is less valuable as the posterior belief of the model parameters will be more diffuse and similar to the prior. As parties prioritize obtaining a model for future predictions and may face legal/decision difficulties in implementing monetary payments, we reward each party with *posterior samples of the model parameters* (in short, *model reward*) instead.

**How should the reward scheme be designed to satisfy the aforementioned privacy, individual rationality, and fairness incentives (Sec. 4)?** Our scheme will naturally satisfy the privacy incentive as any post-processing of the perturbed SS will preserve DP. To satisfy fairness and individual rationality, we set the *target* reward value for every party using $\rho$-Shapley value [47]. **Lastly, to realize these target reward values, how should the model reward be generated for each party (Sec. 5)?** Instead of rewarding all parties with samples from the *same* (grand coalition's) posterior of the model parameters given all their perturbed SS (which would be unfair if their valuations differ), our reward control mechanism generates a *different* posterior for each party that still *preserves a high similarity to the grand coalition's posterior*. Concretely, the mediator scales the SS by a factor between 0 and 1 before sampling to control the impact of data on the posterior (by *tempering* the data likelihood). Scaling the SS by a factor of 0, 1, and between 0 and 1 yield the prior, posterior, and their interpolation, respectively. We then solve for the factor to achieve the target reward value.

By answering the above questions, our work here provides the following novel contributions[3]:

- A new *privacy-valuation trade-off* criterion for valuation functions that is provably satisfied by the combination of our *Bayesian surprise* valuation function with DP noise-aware inference (Sec. 3);
- New incentives including *DP* (while *deterring excessive DP*) and *similarity to grand coalition's model* (Sec. 4);

---

[2]The work of [30] describes problems posed by excessive data privacy and "the need to balance privacy with fuller and representative data collection" (instead of privileging privacy). But, parties are still free to seek DP.

[3]See App. B for the key differences of our work here vs. data valuation and DP/FL works.

- *Reward control mechanisms* (Sec. 5) to generate *posterior samples* of the model parameters for each party that achieve a target reward value and the aforementioned incentives; one such mechanism tempers the likelihood of the data by scaling the SS and data quantity.

## 2 Collaborative ML Problem with Privacy Incentive

Our private collaborative ML problem setup comprises a mediator coordinating information sharing, valuation, and reward, and $n$ parties performing a common ML task (e.g., predicting disease progression). Let the set $N \triangleq \{1, \ldots, n\}$ denote the grand coalition of $n$ parties. Each party $i$ owns a private dataset $\mathcal{D}_i$ which cannot be directly shared with others, including the mediator. *What alternative information should each party provide to the mediator for collaborative training of an ML model?*

To ease aggregation, this work focuses only on Bayesian models with *sufficient statistic* (SS), such as exponential family models [5], Bayesian linear regression [39], and generalized linear models, including Bayesian logistic regression [21] (with approximate SS).

**Definition 2.1** (Sufficient Statistic (SS) [48, 52]). The statistic $s_i$ is a SS for the dataset $\mathcal{D}_i$ if the model parameters $\theta$ and dataset $\mathcal{D}_i$ are conditionally independent given $s_i$, i.e., $p(\theta|s_i, \mathcal{D}_i) = p(\theta|s_i)$.

We propose that each party $i$ shares its SS $s_i$ for and *in place of* its dataset $\mathcal{D}_i$ to protect the privacy of $\mathcal{D}_i$. We assume that the parties have agreed to adopt a common Bayesian model with the same prior $p(\theta)$ of model parameters $\theta$, and each party $i$'s dataset $\mathcal{D}_i$ is independently drawn from the likelihood $p(\mathcal{D}_i|\theta)$ that is conjugate to the prior $p(\theta)$ (i.e., belonging to an exponential family). The mediator can compute the posterior belief $p(\theta|\{\mathcal{D}_i\}_{i \in N})$ of model parameters $\theta$ given the grand coalition $N$'s datasets using a function $f_\theta$ of the sum over shared SS: $p(\theta|\{\mathcal{D}_i\}_{i \in N}) \propto p(\theta) \, f_\theta(\sum_{i \in N} s_i)$. We give a concrete example and the mathematical details of SS in Apps. A.1 and E, respectively.

**Privacy Incentive.** However, sharing the exact SS $s_N \triangleq \{s_i\}_{i \in N}$ will not ensure privacy as the mediator can draw inferences about individual datum in the private datasets $\mathcal{D}_N \triangleq \{\mathcal{D}_i\}_{i \in N}$. To mitigate the privacy risk, each party $i$ should choose its required privacy level $\epsilon_i$ and enforce $(\lambda, \epsilon_i)$-Rényi *differential privacy*.[4] In Def. 2.2, a smaller $\epsilon_i$ corresponds to a stronger DP guarantee.[4]

**Definition 2.2** (Rényi Differential Privacy (DP) [38]). A randomized algorithm $\mathcal{R} : \mathcal{D} \to o$ is $(\lambda, \epsilon)$-Rényi differentially private if for all neighboring datasets $\mathcal{D}$ and $\mathcal{D}'$, the Rényi divergence[4] of order[4] $\lambda > 1$ is $D_\lambda(\mathcal{R}(\mathcal{D}) \, || \, \mathcal{R}(\mathcal{D}')) \leq \epsilon$.

Party $i$ can enforce (example-level)[4] $(\lambda, \epsilon_i)$-Rényi DP by applying the Gaussian mechanism: It generates perturbed SS $o_i \triangleq s_i + z_i$ by sampling a Gaussian noise vector $z_i$ from the distribution $p(Z_i) = \mathcal{N}(\mathbf{0}, \, 0.5 \, (\lambda/\epsilon_i) \, \Delta_2^2(g) \, \mathbf{I})$ where $\Delta_2^2(g)$ is the squared $\ell_2$-sensitivity[4] of the function $g$ that maps the dataset $\mathcal{D}_i$ to the SS $s_i$. We choose Rényi DP over the commonly used $(\epsilon, \delta)$-DP as it gives a stronger privacy definition and allows a more convenient composition of the Gaussian mechanisms [38], as explained in App. A.2.

Each party $i$ will share (i) the number $c_i \triangleq |\mathcal{D}_i|$ of data points in its dataset $\mathcal{D}_i$, (ii) its perturbed SS $o_i$,[5] and (iii) its Gaussian distribution $p(Z_i)$ with the mediator. As DP algorithms are robust to post-processing, the mediator's subsequent operations of $o_i$ (with no further access to the dataset) will preserve the same DP guarantees. The mediator uses such information to quantify the impact of the DP noise and compute the DP noise-aware posterior[6] $p(\theta|\{o_i\}_{i \in N})$ via *Markov Chain Monte Carlo* (MCMC) sampling steps outlined by [3, 4, 27].

In this section, we have satisfied the privacy incentive. In Sec. 3, we assign a value $v_C$ to each coalition $C \subseteq N$'s perturbed SS $o_C \triangleq \{o_i\}_{i \in C}$ that would decrease, on average, as the DP guarantee strengthens. In Secs. 4 and 5, we outline our reward scheme: Each party $i$ will be rewarded with model parameters sampled from $q_i(\theta)$ (in short, *model reward*) for future predictions with an appropriate reward value $r_i$ (decided based on $(v_C)_{C \subseteq N}$) to satisfy collaborative ML incentives (e.g., individual

---

[4] Parties agree on $\lambda$. In App. A.2, we will define/explain the DP-related concepts and other DP notions.

[5] A benefit of sharing perturbed SS is that each party only incurs the privacy cost *once* regardless of the number of samples drawn or coalitions considered.

[6] Here, $o_i$ should be interpreted as a random vector taking the value of its sample. Non-noise-aware inference incorrectly treats $o_i$ as $s_i$ and computes $p(\theta|\{s_i = o_i\}_{i \in N})$ instead. See App. A.3.

rationality, fairness). Our work's main contributions, notations, and setup are detailed in Fig. 1. The main steps involved are detailed in Algo. 2.

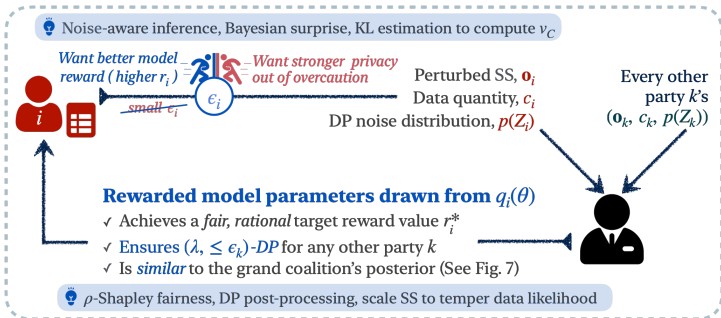

Figure 1: An overview of our private collaborative ML problem setup from party $i$'s perspective and our novel contributions (ideas in blue, novel combination of solutions in blue). We (i) enforce a privacy-reward trade-off (using each party $i$'s desire for a higher-quality model reward in collaborative ML) to deter party $i$ from unfetteredly/overcautiously selecting an excessive DP guarantee (small $\epsilon_i$), (ii) ensure DP in valuation and rewards, and (iii) preserve similarity of its model reward $q_i(\theta)$ to the grand coalition $N$'s posterior $p(\theta|\boldsymbol{o}_N)$ to achieve a high utility.

## 3 Valuation of Perturbed Sufficient Statistics

The perturbed SS $\boldsymbol{o}_C$ of coalition $C$ is more valuable and assigned a higher value $v_C$ if it yields a model (in our work here, the DP noise-aware posterior $p(\theta|\boldsymbol{o}_C)$) of higher quality. Most data valuation methods [24, 18, 63] measure the quality of an ML model by its performance on a validation set. However, it may be challenging for collaborating parties (e.g., competing healthcare firms) to create and agree on a large, representative validation set as they may prioritize accurate predictions on different inputs (e.g., patient demographics) [47]. The challenge increases when each firm requires privacy and avoids data sharing. Other valuation methods [47, 59] have directly used the private inputs of the data (e.g., design matrix). Here, we propose to value the perturbed SS $\boldsymbol{o}_C$ of coalition $C$ based on the *surprise* [23] that it elicits from the prior belief of model parameters, as defined below:

**Definition 3.1** (Valuation via Bayesian Surprise). The value of coalition $C$ or its *surprise* $v_C$ is the KL divergence $D_{\mathrm{KL}}(p(\theta|\boldsymbol{o}_C); p(\theta))$ between posterior $p(\theta|\boldsymbol{o}_C)$ vs. prior $p(\theta)$.

From Def. 3.1, a greater surprise would mean that more bits will be needed to encode the information in $p(\theta|\boldsymbol{o}_C)$ given that others already know $p(\theta)$. Otherwise, a smaller surprise means our prior belief has not been updated significantly. Moreover, as the valuation depends on the observed $\boldsymbol{o}_C$, the surprise elicited by the exact SS and data points will indirectly influence the valuation. Next, by exploiting the equality of the expected Bayesian surprise and the information gain on model parameters $\theta$ given perturbed SS $\boldsymbol{o}_C$ (i.e., $\mathbb{E}_{\boldsymbol{o}_C}[v_C] = \mathbb{I}(\theta; \boldsymbol{o}_C)$), we can establish the following essential properties of our valuation function:

V1 **Non-negativity.** $\forall C \subseteq N \; \forall \boldsymbol{o}_C \;\; v_C \geq 0$ . This is due to the non-negativity of KL divergence.

V2 **Party monotonicity.** In expectation w.r.t. $\boldsymbol{o}_C$, adding a party will not decrease the valuation: $\forall C \subseteq C' \subseteq N \;\; \mathbb{E}_{\boldsymbol{o}_{C'}}[v_{C'}] \geq \mathbb{E}_{\boldsymbol{o}_C}[v_C]$ . The proof (App. C.1) uses the "information never hurts" property.

V3 **Privacy-valuation trade-off.** When the DP guarantee is strengthened from $\epsilon_i$ to a smaller $\epsilon_i^s$ and independent Gaussian noise is added to $\boldsymbol{o}_i$ to generate $\boldsymbol{o}_i^s$, in expectation, the value of any coalition $C$ containing $i$ will strictly decrease: Let $v_C^s$ denote the value of coalition $C$ with the random variable and realization of $\boldsymbol{o}_i$ replaced by $\boldsymbol{o}_i^s$. Then, $(i \in C) \wedge (\epsilon_i^s < \epsilon_i) \Rightarrow \mathbb{E}_{\boldsymbol{o}_C}[v_C] > \mathbb{E}_{\boldsymbol{o}_C^s}[v_C^s]$ .

The proof of V3 (App. C.1) uses the data processing inequality of information gain and the conditional independence between $\theta$ and $\boldsymbol{o}_i^s$ given $\boldsymbol{o}_i$. Together, these properties address an important question of **how to ensure DP and value a party's data to enforce a privacy-valuation trade-off** (Sec. 1). Additionally, in App. C.2, we prove that in expectation, our Bayesian surprise valuation is equivalent to the alternative valuation that measures the similarity of $p(\theta|\boldsymbol{o}_C)$ to the grand coalition $N$'s DP noise-aware posterior $p(\theta|\boldsymbol{o}_N)$.

**Implementation.** Computing the Bayesian surprise valuation is intractable since the DP noise-aware posterior $p(\theta|\boldsymbol{o}_C)$ and its KL divergence from $p(\theta)$ do not have a closed-form expression. Nonetheless, there exist approximate inference methods like the *Markov chain Monte Carlo* (MCMC) sampling to estimate $p(\theta|\boldsymbol{o}_C)$ efficiently, as discussed in App. A.3. As our valuation function requires estimating the value of multiple coalitions and the posterior sampling step is costly, we prefer estimators with a low time complexity and a reasonable accuracy for a moderate number $m$ of samples. We recommend KL estimation to be performed using the nearest-neighbors method [45], and repeated and averaged to reduce the variance of the estimate (see App. C.3 for a discussion). The nearest-neighbor KL estimator is also asymptotically unbiased; drawing more samples would reduce the bias and variance of our estimates and is more likely to ensure fairness — for example, party $i$'s *sampled* valuation is only larger than $j$'s if $i$'s *true* valuation is higher.

**Remark.** Our valuation is based on the submitted information $\{c_i, \boldsymbol{o}_i, p(Z_i)\}_{i \in N}$ without verifying or incentivizing their truthfulness. We discuss how this limitation is shared by existing works and can be overcome by legal contracts and trusted data-sharing platforms in App. I.

## 4 Reward Scheme for Ensuring Incentives

After valuation, the mediator should reward each party $i$ with a *model reward* (i.e., consisting of samples from $q_i(\theta)$) for future predictions. Concretely, $q_i(\theta)$ is a belief of model parameters $\theta$ after learning from the perturbed SS $\boldsymbol{o}_N$. As in Sec. 3, we value party $i$'s model reward as the KL divergence from the prior: $r_i \triangleq D_{\mathrm{KL}}(q_i(\theta); p(\theta))$. The mediator will first *decide* the target reward value $r_i^*$ for every party $i \in N$ using $\{v_C\}_{C \subseteq N}$ to satisfy incentives such as fairness. The mediator will then *control* and generate a *different* $q_i(\theta)$ for every party $i \in N$ such that $r_i = r_i^*$ using reward control mechanisms from Sec. 5. We will now outline the incentives and desiderata for model reward $q_i(\theta)$ and reward values $r_i$ and $r_i^*$ for every party $i \in N$ when the grand coalition forms[7].

P1 **DP-Feasibility.** In party $i$'s reward, any other party $k$ is still guaranteed at least its original $(\lambda, \epsilon_k)$-DP guarantee or stronger. The implication is that the generation of party $i$'s reward should not require more private information (e.g., SS) from party $k$.

P2 **Efficiency.** There is a party $i \in N$ whose model reward is the grand coalition $N$'s posterior, i.e., $q_i(\theta) = p(\theta|\boldsymbol{o}_N)$. It follows that $r_i = v_N$ .

P3 **Fairness.** The target reward values $(r_i^*)_{i \in N}$ must consider the coalition values $\{v_C\}_{C \subseteq N}$ and satisfy properties F1 to F4 given in [47] and reproduced in App. D.2. The monotonicity axiom F4 ensures using a valuation function which enforces that a *privacy-valuation trade-off* will translate to a *privacy-reward trade-off* and deter parties from selecting excessive DP guarantees.

P4 **Individual Rationality.** Each party should receive a model reward that is more valuable than the model trained on its perturbed SS alone: $\forall i \in N \;\; r_i^* \geq v_i$ .

P5 **Similarity to Grand Coalition's Model.** Among multiple model rewards $q_i(\theta)$ whose value $r_i$ equates the target reward $r_i^*$, we *secondarily* prefer one with a higher similarity $r_i' = -D_{\mathrm{KL}}(p(\theta|\boldsymbol{o}_N); q_i(\theta))$ to $p(\theta|\boldsymbol{o}_N)$.[8]

P6 **Group Welfare.** The reward scheme should maximize the total reward value $\sum_{i=1}^n r_i$ to increase the utility of model reward for each party and achieve the aims of collaborative ML.

**Choice of desiderata.** We adopt the desiderata from [47] but make P1 and P2 more specific (by considering each party's actual reward $q_i(\theta)$ over just its values $r_i$ and $v_N$) and introduce P5. Firstly, for our Bayesian surprise valuation function, the feasibility constraint of [47] is inappropriate as removing a party or adding some noise realization may result in $r_i > v_N$,[9] so we propose P1 instead. Next, we recognize that party $i$ is not indifferent to all model rewards $q_i(\theta)$ with the same target reward value as they may have different utility (predictive performance). Thus, we propose our more specific P2 and a secondary desideratum P5. As P5 is considered after other desiderata, it does not conflict with existing results, e.g., design for $(r_i^*)_{i \in N}$ to satisfy other incentives.

*Remark on Rationality.* In P4, a party's model reward is compared to the model trained its *perturbed* SS instead of its *exact* SS alone. This is because the mediator cannot access (and value the model

---

[7]See App. D.1 for the modifications needed when the grand coalition does not form.

[8]Expectation propagation, a common approximate inference technique, also maximizes $r_i'$. [5].

[9]The valuation fn. is only monotonic in expectation (V2-V3). See further discussion in App. I, Question 9.

trained on) the private exact SS. Moreover, with no restrictions on the maximum DP noise, the value of some party's exact SS may exceed the grand coalition's perturbed SS when parties require strong DP guarantees. P4 is sufficient when parties require DP when alone to protect data from curious users of their ML model [1, 3, 27]. For example, a hospital may not want doctors to infer specific patients' data. When parties do not require DP when alone, our reward scheme cannot theoretically ensure that the model reward from collaboration is better than using the exact SS. We further discuss this limitation in App. D.3.

**Design of** $(r_i^*)_{i \in N}$**.** To satisfy the desiderata from [47] (including our fairness P3 and rationality P4 incentives), we adopt their $\rho$-Shapley fair reward scheme with $\rho \in (0, 1]$ that sets $r_i^* = v_N(\phi_i / \max_{k \in N} \phi_k)^\rho$ with Shapley value[10] $\phi_i \triangleq (1/n) \sum_{C \subseteq N \setminus i} \left[ \binom{n-1}{|C|}^{-1} \left( v_{C \cup \{i\}} - v_C \right) \right]$. Shapley value's consideration of *marginal contribution* (MC) to *all* coalitions is key to ensuring strict desirability F3 such that party $i$ obtains a higher reward than party $k$ (despite $v_i = v_k$) if $i$'s perturbed SS adds more value to every other non-empty coalition. Applying Theorem 1 of [47], the mediator should set $\rho$ between 0 and $\min_{i \in N} \log(v_i/v_N)/\log(\phi_i/\max_k \phi_k)$ to guarantee rationality. Selecting a larger $\rho$ incentivizes a party with a high-quality perturbed SS to share by fairly limiting the benefits to parties with lower-quality ones. Selecting a smaller $\rho$ reward parties more equally and increase group welfare P6. Refer to Sec. 4.2 of [47] for a deeper analysis of the impact of varying $\rho$. These results hold for any choice of $(\lambda, \epsilon_i)$.

After explaining the desiderata for model reward $q_i(\theta)$ and reward values $r_i$ and $r_i^*$ for every party $i \in N$, we are now ready to solve for $q_i(\theta)$ such that $r_i = r_i^*$.

# 5 Reward Control Mechanisms

This section discusses two mechanisms to generate model reward $q_i(\theta)$ with different attained reward value $r_i$ for every party $i \in N$ by controlling a *single* continuous parameter and solving for its value such that the attained reward value equates the target reward value: $r_i = r_i^*$. We will discuss the more obvious reward mechanism in Sec. 5.1 to contrast its cons with the pros of that in Sec. 5.2. Both reward mechanisms do not request new information from the parties; thus, the DP post-processing property applies, and every party $k$ is still guaranteed at least its original DP guarantee or stronger in all model rewards (i.e., P1 holds).

## 5.1 Reward Control via Noise Addition

The work of [47] controls the reward values by adding Gaussian noise to the data outputs. We adapt it such that the mediator controls the reward value for party $i \in N$ by adding Gaussian noise to the perturbed SS of each party $k \in N$ instead. To generate the model reward for party $i$ (superscripted), the mediator will reparameterize the sampled Gaussian noise vectors $\{\boldsymbol{e}_k^i \sim \mathcal{N}(\boldsymbol{0}, \boldsymbol{I})\}_{k \in N}$ to generate the further perturbed SS[11]

$$\boldsymbol{t}_N^i \triangleq \left\{ \boldsymbol{t}_k^i \triangleq \boldsymbol{o}_k + \left( 0.5 \, \lambda \, \Delta_2^2(g_k) \, \tau_i \right)^{1/2} \boldsymbol{e}_k^i \right\}_{k \in N}$$

where $\Delta_2^2(g_k)$ is the squared $\ell_2$-sensitivity of function $g_k$ that computes the exact SS $\boldsymbol{s}_k$ from dataset $\mathcal{D}_k$ (Sec. 2). Then, the mediator rewards party $i$ with samples of model parameters $\theta$ from the new DP noise-aware posterior $q_i(\theta) = p(\theta | \boldsymbol{t}_N^i)$.

Here, the scalar $\tau_i \geq 0$ controls the additional noise variance and can be optimized via root-finding to achieve $r_i = r_i^*$. The main advantage of this reward control mechanism is its interpretation of strengthening party $k$'s DP guarantee in all parties' model rewards (see P1). For example, it can be derived that if $\epsilon_k = \infty$, then party $k$ will now enjoy $(\lambda, 1/\tau_i)$-DP guarantee in party $i$'s reward instead. If $\epsilon_k < \infty$, then party $k$ will now enjoy a stronger $(\lambda, \epsilon_k/(1 + \tau_i \epsilon_k))$-DP guarantee since $\epsilon_k/(1 + \tau_i \epsilon_k) < \epsilon_k$.

However, this mechanism has some disadvantages. Firstly, for the same scaled additional noise variance $\tau_i$, using different noise realizations $\{\boldsymbol{e}_k^i\}_{k \in N}$ will lead to model reward $q_i(\theta)$ with varying

---

[10] Party $i$'s MC to some coalitions and Shapley value $\phi_i$ may be negative, which results in an unusable negative/undefined $r_i^*$. This issue can be averted while preserving P3 by upweighting non-negative MCs such as to the empty set.

[11] To ease notation, we slightly abuse $\boldsymbol{t}_k^i$ to represent both a random vector and its sample.

similarity $r_i'$ to the grand coalition $N$'s posterior. The mechanism cannot efficiently select the best model reward with higher $r_i'$ (P5). Secondly, the value of $r_i$ computed using such $\boldsymbol{t}_k^i$ may be non-monotonic[12] in $\tau_i$ (see Fig. 2d), which makes it hard to bracket the smallest root $\tau_i$ that solves for $r_i = r_i^*$. To address these disadvantages, we will propose the next mechanism.

## 5.2 Reward Control via Likelihood Tempering

Intuitively, a party $i$ who is assigned a lower target reward value $r_i^* < v_N$ should be rewarded with posterior samples of model parameters $\theta$ that use *less* information from the datasets and SS of all parties. Sparked by the diffuse posterior algorithm [16], we propose that the mediator can generate such "less informative" samples for party $i$ using the *normalized* posterior[13]

$$q_i(\theta) \propto p(\theta) \left[ p(\mathcal{D}_N | \theta) \right]^{\kappa_i} \tag{1}$$

involving the product of the prior $p(\theta)$ and the data likelihood $p(\mathcal{D}_N | \theta)$ to the power of (or, said in another way, tempered by a factor of) $\kappa_i$. Notice that setting $\kappa_i = 0$ and $\kappa_i = 1$ recover the prior $p(\theta)$ and the posterior $p(\theta | \mathcal{D}_N)$, respectively. Thus, setting $\kappa_i \in (0, 1)$ should smoothly interpolate between both. We can optimize $\kappa_i$ to control $q_i(\theta)$ so that $r_i = r_i^* < v_N$.

But, *how do we temper the likelihood?* We start by examining the easier, non-private setting. In Sec. 2, we stated that under our assumptions, the posterior $p(\theta | \mathcal{D}_N)$ can be computed by using the sum of data quantities $\{c_k\}_{k \in N}$ and sum of exact SS $\boldsymbol{s}_N$. In App. E, we further show that using the tempered likelihood $[p(\mathcal{D}_N | \theta)]^{\kappa_i}$ is equivalent to scaling the data quantities and the exact SS $\boldsymbol{s}_N$ by the factor $\kappa_i$ beforehand. In the private setting, the mediator can similarly scale the data quantities, the perturbed SS in $\boldsymbol{o}_N$ (instead of the inaccessible exact SS), and the $\ell_2$-sensitivity by the factor $\kappa_i$ beforehand; see App. E.3 for details. This likelihood tempering mechanism addresses both disadvantages of Sec. 5.1:

- There is no need to sample additional DP noise. We empirically show that tempering the likelihood produces a model reward that *interpolates* between the prior vs. posterior (in App. G) and *preserves* a higher similarity $r_i'$ to the grand coalition $N$'s posterior (P5 and hence, more group welfare P6) and better predictive performance than noise addition (see Sec. 6).
- Using a smaller tempering factor $\kappa_i \in [0, 1]$ provably decreases the attained reward value $r_i$ (see App. E). Thus, as the relationship between $r_i$ and $\kappa_i$ is monotonic, we can find the only root by searching the interval $[0, 1]$.

**Remark.** Our discussion on improving the estimate of $v_C$ in the paragraph on implementation in Sec. 3 also applies to the estimate of $r_i$ in Secs. 5.1 and 5.2. Thus, solving for $\tau_i$ or $\kappa_i$ to achieve $r_i = r_i^*$ using any root-finding algorithm can only be accurate up to the variance in our estimate.

## 6 Experiments and Discussion

This section empirically evaluates the privacy-valuation and privacy-reward trade-offs (Sec. 6.1), reward control mechanisms (Sec. 6.2), and their relationship with the utility of the model rewards (Sec. 6.3). The time complexity of our scheme is analyzed in App. F and baseline methods are discussed in App. H.3. We consider *Bayesian linear regression* (BLR) with unknown variance on the Syn and CaliH datasets, and *Bayesian logistic regression* on the Diab dataset with 3 collaborating parties (see App. H.1 for details) and enforce $(2, \epsilon_i)$-Rényi DP. For **Synthetic BLR (Syn)**, we select and use a *normal inverse-gamma* distribution (i) to generate the true regression model weights, variance, and a 2D-dataset and (ii) as our model prior $p(\theta)$. We consider 3 parties with $c_1 = 100$, $c_2 = 200$, $c_3 = 400$ data points, respectively. For **Californian Housing dataset (CaliH)** [44], as in [47], 60% of the CaliH data is deemed "public/historic" and used to pre-train a neural network without DP. Real estate firms may only care about the privacy of their newest transactions. As the parties' features-house values relationship may differ from the "public" dataset, we do transfer learning and selectively retrain only the last layer with BLR using the parties' data. Parties 1 to 3 have,

---

[12]Increasing $\tau_i$ (i.e., using $\boldsymbol{t}_k^i$ that diverges more from $\boldsymbol{s}_k$) can instead increase the surprise: The privacy-valuation trade-off (Sec. 3) only holds in *expectation* across all noise and SS realizations.

[13]The normalized posterior is also known as the power posterior. [37] discuss useful interpretation and benefits such as synthetically reducing the sample size, increasing the ease of computation/MCMC mixing and robustness to model misspecifications.

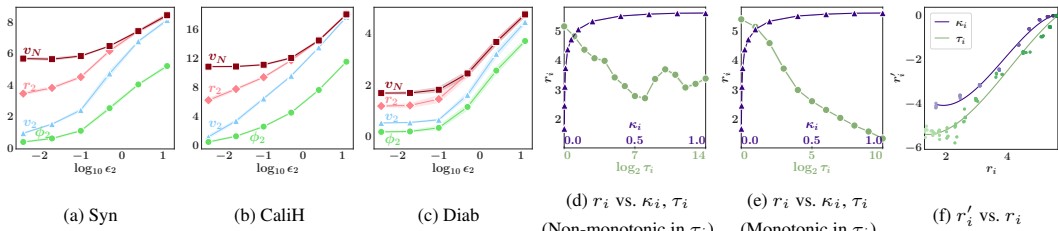

| (a) Syn | (b) CaliH | (c) Diab | (d) $r_i$ vs. $\kappa_i, \tau_i$ (Non-monotonic in $\tau_i$) | (e) $r_i$ vs. $\kappa_i, \tau_i$ (Monotonic in $\tau_i$) | (f) $r'_i$ vs. $r_i$ |

Figure 2: (a-c) Graphs of party 2's valuation $v_2$, Shapley value $\phi_2$, attained reward value $r_2$ vs. privacy guarantee $\epsilon_2$ for various datasets. (d-e) Graphs of attained reward value $r_i$ vs. $\kappa_i$ (Sec. 5.2) and $\tau_i$ (Sec. 5.1) for 2 different noise realizations. (f) Graph of similarity $r'_i$ to grand coalition $N$'s posterior $p(\theta|\boldsymbol{o}_N)$ vs. $r_i$ for Syn dataset corresponding to (e).

respectively, $20\%, 30\%$, and $50\%$ of the dataset with $6581$ data points and $6$ features. For **PIMA Indian Diabetes classification dataset (Diab)** [50], we use a Bayesian logistic regression model to predict whether a patient has diabetes based on sensitive inputs (e.g., patient's age, BMI, number of pregnancies). To reduce the training time, we only use the $4$ PCA main components as features (to generate the approximate SS) [27]. Parties $1, 2$, and $3$ have, respectively, $20\%, 30\%$, and $50\%$ of the dataset with $614$ data points. As we are mainly interested[14] in the impact of one party controlling its privacy guarantee $\epsilon_i$, for all experiments, we only vary party 2's from the default $0.1$. We fix the privacy guarantees of others ($\epsilon_1 = \epsilon_3 = 0.2$) and $\rho = 0.2$ in the $\rho$-Shapley fair reward scheme, and analyze party 2's reward and utility. Note that as $\epsilon_2$ increases (decreases), party 2 becomes the most (least) valuable of all parties.

## 6.1 Privacy-valuation and Privacy-reward Trade-offs

For each dataset, we only vary the privacy guarantee of party $i = 2$ with $\epsilon_2 \in [0.004, 0.02, 0.1, 0.5, 2.5, 12.5]$ and use the Gaussian mechanism and a fixed random seed to generate the perturbed SS $\boldsymbol{o}_2$ from the exact SS $\boldsymbol{s}_2$. Fig. 2a-c plot the mean and shades the standard error of $v_i, v_N, \phi_i$, and $r_i$ over $5$ runs. The privacy-valuation and privacy-reward trade-offs can be observed: As the privacy guarantee weakens (i.e., $\epsilon_2$ increases), party 2's valuation $v_2$, Shapley value $\phi_2$, and attained reward value $r_2$ increase. When $\epsilon_2$ is large, party 2 will be the most valuable contributor and rewarded with $p(\theta|\boldsymbol{o}_N)$, hence attaining $r_i = v_N$. App. H.5 shows that the trade-offs do not hold for non-noise-aware inference.

## 6.2 Reward Control Mechanisms

We use the Syn experiment to compare the reward mechanisms that vary the noise addition using $\tau_i$ (Sec. 5.1) vs. temper the likelihood using $\kappa_i$ (Sec. 5.2). The mechanisms control $q_i(\theta)$ (i.e., used to generate party $i$'s model reward) to attain the target reward values. For each value of $\tau_i$ and $\kappa_i$ considered, we repeat the posterior sampling and KL estimation method $5$ times. Figs. 2d and 2e-f use different sets of sampled noise $\{\boldsymbol{e}_k^i\}_{k \in N}$ to demonstrate the stochastic relationship between $r_i$ and $\tau_i$. In Fig. 2d, the non-monotonic disadvantage of noise addition can be observed: As $\tau_i$ increases, $r_i$ does not consistently decrease, hence making it hard to solve for the smaller $\tau_i$ that attains $r_i^* = 3$. In contrast, as $\kappa_i$ decreases from $1$, $r_i$ consistently decreases. Furthermore, in Fig. 2f, we demonstrate the other advantage of likelihood tempering: For the same $r_i$, tempering the likelihood leads to a higher similarity $r'_i$ to the posterior $p(\theta|\boldsymbol{o}_N)$ than noise addition. In App. H.6, we report the relationship between $r_i$ vs. $\kappa_i$ and $\tau_i$ for the other real-world datasets.

## 6.3 Utility of Model Reward

The utility (or the predictive performance) of both Bayesian models can be assessed by the *mean negative log probability* (MNLP) of a non-private test set.[15] In short, MNLP reflects how unlikely the test set is given the perturbed SS and additionally cares about the uncertainty/confidence in the model predictions (e.g., due to the impact of DP noise). MNLP will be higher (i.e., worse) when the model

---

[14]We defer the analysis of valuation function (e.g., impact of varying coverage of input space) to App. H.4.

[15]Such a test set is hard to obtain in practice and we are only using it for evaluation purposes.

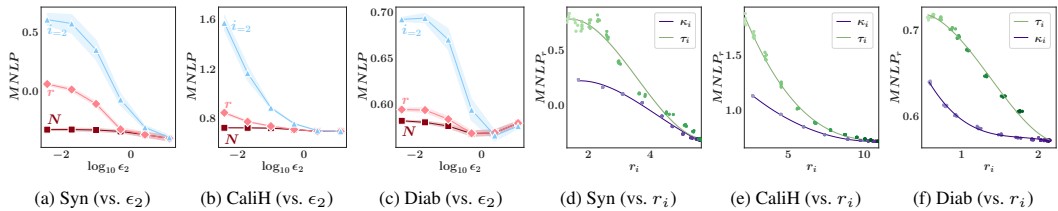

| (a) Syn (vs. $\epsilon_2$) | (b) CaliH (vs. $\epsilon_2$) | (c) Diab (vs. $\epsilon_2$) | (d) Syn (vs. $r_i$) | (e) CaliH (vs. $r_i$) | (f) Diab (vs. $r_i$) |

Figure 3: (a-c) Graphs of utility of party 2's model reward measured by $\mathrm{MNLP}_r$ vs. privacy guarantee $\epsilon_2$ for various datasets. (d-f) Graphs of utility of model reward measured by $\mathrm{MNLP}_r$ vs. attained reward value $r_i$ under the two reward control mechanisms for various datasets.

is more uncertain of its accurate predictions or overconfident in inaccurate predictions on the test set; see App. H.2 for an in-depth definition.

**Privacy trade-offs.** Figs. 3a-c illustrate the privacy-utility trade-off described in Sec. 1: As $\epsilon_2$ decreases (i.e., privacy guarantee strengthens), the $\mathrm{MNLP}_N$ of grand coalition $N$'s collaboratively trained model and the $\mathrm{MNLP}_i$ of party $i = 2$'s individually trained model generally increase, so their utilities drop (†). This motivates the need to incentivize party 2 against selecting an excessively small $\epsilon_2$ by enforcing privacy-valuation and privacy-reward trade-offs. From Figs. 3a-c, the impact of our scheme *enforcing* the trade-offs can be observed: As $\epsilon_2$ decreases, the $\mathrm{MNLP}_r$ of party $i = 2$'s model reward increases.

*Remark.* In Fig. 3c, an exception to (†) is observed. The exception illustrates that the privacy-valuation trade-off may not hold for a valuation function based on the performance on a validation set.

**Individual rationality.** It can be observed from Figs. 3a-c that as $\epsilon_2$ decreases, the $\mathrm{MNLP}_r$ of party $i = 2$'s model reward increases much less rapidly than the $\mathrm{MNLP}_i$ of its individually trained model. So, it is rational for party $i = 2$ to join the collaboration to get a higher utility.

*Remark.* Party $i = 2$'s utility gain appears small when $\epsilon_2$ is large due to parties 1 and 3's selection of strong privacy guarantee $\epsilon = 0.2$. Party $i$ can gain more when other parties require weaker privacy guarantees such as $\epsilon = 2$ instead (see App. H.5).

**Likelihood tempering is a better reward control mechanism.** Extending Sec. 6.2, we compare the utility of party $i$'s model reward generated by noise addition vs. likelihood tempering in Figs. 3d-f. Across all experiments, likelihood tempering (with $\kappa_i$) gives (i) a lower $\mathrm{MNLP}_r$ and hence a higher utility, and (ii) a lower variance in $\mathrm{MNLP}_r$ than varying the noise addition (with $\tau_i$).

## 7 Related Works

Fig. 5 in App. B gives a diagrammatic overview showing how our work fills the gap in existing works.

**Data Valuation.** Most data valuation methods are not differentially private and directly access the data. For example, computing the information gain [47] or volume [59] requires the design matrix. While it is possible to make these valuation methods differentially private (see App. H.3) or value DP trained models using validation accuracy (on an agreed, *public* validation set), the essential properties of our valuation function (V2-V3) may not hold.

**Privacy Incentive.** Though the works of [20, 60] reward parties directly proportional to their privacy budget, their methods do not incentivize data sharing as a party does not fairly receive a higher reward value for contributing a larger, more informative dataset. While the work of [28] artificially creates a privacy-reward trade-off by paying each party $i$ the product of its raw data's Shapley value $\phi_i$ and a monotonic transformation of $\epsilon_i$, it neither ensures DP w.r.t. the mediator nor fairly considers how a stronger DP guarantee may reduce party $i$'s marginal contribution to others (hence $\phi_i$). The work of [13] considers data acquisition from parties with varying privacy requirements but focuses on the mean estimation problem and designing payment and privacy loss functions to get parties to report their true unit cost of privacy loss. Our work here distinctively considers Bayesian models and fairness and allows parties to choose their privacy guarantees directly while explicitly enforcing a privacy-reward trade-off.

**Difficulties ensuring incentives with existing DP/FL works.** The *one posterior sampling* (OPS) method [56, 16] proposes that each party $i$ can achieve DP by directly releasing samples from the posterior $p(\theta|\mathcal{D}_i)$ (if the log-likelihood is bounded). However, OPS is data inefficient and may not guarantee privacy for approximate inference [15]. It is unclear how we can privately value a coalition $C$ and sample from the joint posterior $p(\theta|\{\mathcal{D}_i\}_{i\in C})$. `DP-FedAvg/DP-FedSGD` [36] or `DP-SGD` [1] enable collaborative but private training of neural networks by requiring each party $i$ to clip and add Gaussian noise to its submitted gradient updates. However (in addition to the valuation function issue above), it is tricky to ensure that the parties' rewards satisfy data sharing incentives. In each round of FL, parties selected will receive the (same) latest model parameters to compute gradient updates. This setup goes against the fairness (P3) incentive as parties who share less informative gradients should be rewarded with lower quality model parameters instead. Although the unfairness may potentially be corrected via gradient-based [58] or monetary rewards, there is no DP reward scheme to guarantee a party better model reward from collaboration than in solitude or a higher monetary reward than its cost of participation, hence violating individual rationality.

# 8  Conclusion

Unlike existing works in collaborative ML that solely focus on the fairness incentive, our proposed scheme further (i) ensures privacy for the parties during valuation and in model rewards and (ii) enforces a privacy-valuation trade-off to deter parties from unfetteredly selecting excessive DP guarantees to maximize the utility of collaboratively trained models.[16] This involves novelly combining our proposed Bayesian surprise valuation function and reward control mechanism with DP noise-aware inference. We empirically evaluate our scheme on several datasets. Our likelihood tempering reward control mechanism consistently preserves better predictive performance.

Our work has two limitations which future work should overcome. Firstly, we only consider ML models with SS (see App. A.1 for applications) and a single round of sharing information with the mediator as a case study to show the incentives and trade-offs can be achieved. Future work should generalize our scheme to ML models without an explicit SS.

Next, like the works of [18, 17, 25, 40, 47, 51] and others, we do not consider the truthfulness of submitted information and value data *as-is*. This limitation is acceptable for two reasons. 1) Parties such as hospitals and firms will truthfully share information as they are primarily interested in building and receiving a model reward of high quality and may additionally be bound by the collaboration's legal contracts and trusted data-sharing platforms. For example, with the use of X-road ecosystem,[17] parties can upload a private database which the mediator can query for the perturbed SS. This ensures the authenticity of the data (also used by the owner) and truthful computation given the uploaded private database. 2) Each party would be more inclined to submit true information as any party $k$ who submits fake SS will reduce its utility from the collaboration. This is because party $k$'s submitted SS is used to generate $k$'s model reward and cannot be replaced locally as party $k$ will only receive posterior samples. Future work should seek to verify and incentivize truthfulness.

## Acknowledgments and Disclosure of Funding

This research/project is supported by the National Research Foundation Singapore and DSO National Laboratories under the AI Singapore Programme (AISG Award No: AISG2-RP-2020-018). This work is also supported by the National Natural Science Foundation of China (62206139) and the Major Key Project of PCL (PCL2023AS7-1).

---

[16]App. I discusses the ethical implications and limitations of our work, and other questions a reader may have.

[17]`https://joinup.ec.europa.eu/collection/ict-security/solution/x-road-data-exchange-layer/about`, `https://x-road.global/`

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

# A  Fundamental Concepts

In this section, we will elaborate on concepts from Sec. 2 in more detail.

## A.1  Sufficient Statistics

Bayesian learning involves updating our belief of the likely values of the model parameters $\theta$, captured in the prior $p(\theta)$, to a posterior belief $p(\theta|\mathcal{D}_i) \propto p(\theta) \times p(\mathcal{D}_i|\theta)$. The posterior belief gets more concentrated (around the maximum likelihood estimate) after observing a larger dataset $\mathcal{D}_i$.

The statistic $\boldsymbol{s}_i$ is a SS for $\mathcal{D}_i$ if $\theta$ and $\mathcal{D}_i$ are conditionally independent given $\boldsymbol{s}_i$, i.e., $p(\theta|\mathcal{D}_i) = p(\theta|\boldsymbol{s}_i, \mathcal{D}_i) = p(\theta|\boldsymbol{s}_i)$ [48, 52]. Knowing the dataset $\mathcal{D}_i$ does not provide any extra information about $\theta$ beyond the SS $\boldsymbol{s}_i$. SS exists for exponential family models [5] and Bayesian linear regression [39]. Approximate SS has been proposed by [21] for generalized linear models. For more complex data such as images, we can use pre-trained neural networks like VGG-16 as feature extractors and generate SS from the last hidden layer's outputs.

**Bayesian Linear Regression.**  In linear regression, each datum consists of the input $\boldsymbol{x} \in \mathbb{R}^w$ and the output variable $y \in \mathbb{R}$. Let $\mathcal{D}$ denote the dataset with $c$ data points, and $\boldsymbol{y}$ and $\boldsymbol{X}$ be the corresponding concatenated output vector and design matrix in $\mathbb{R}^{c \times w}$. Bayesian linear regression models the relationship as $\boldsymbol{y} = \boldsymbol{X}\boldsymbol{w} + \mathcal{N}(0, \sigma^2 \boldsymbol{I})$ where the model parameters $\theta$ consists of the weight parameters $\boldsymbol{w} \in \mathbb{R}^w$ and the noise variance $\sigma^2$. The likelihood

$$
\begin{aligned}
p(\boldsymbol{y}|\boldsymbol{X}, \boldsymbol{w}, \sigma^2) &= (2\pi\sigma^2)^{-\frac{c}{2}} \exp\left(-\frac{(\boldsymbol{y} - \boldsymbol{X}\boldsymbol{w})^\top (\boldsymbol{y} - \boldsymbol{X}\boldsymbol{w})}{2\sigma^2}\right) \\
&= (2\pi\sigma^2)^{-\frac{c}{2}} \exp\left[\frac{-1}{2\sigma^2}\boldsymbol{y}^\top \boldsymbol{y} + \frac{1}{\sigma^2}\boldsymbol{w}^\top \boldsymbol{X}^\top \boldsymbol{y} - \frac{1}{2\sigma^2}\boldsymbol{w}^\top \boldsymbol{X}^\top \boldsymbol{X}\boldsymbol{w}\right].
\end{aligned}
$$

only depends on data via the sufficient statistics $\boldsymbol{s} = (\boldsymbol{y}^\top \boldsymbol{y}, \boldsymbol{X}^\top \boldsymbol{y}, \boldsymbol{X}^\top \boldsymbol{X})$. Concretely, when the prior $p(\theta)$ of the weights and variance follow a normal inverse-gamma distribution, $\texttt{NIG}(\boldsymbol{0}, \boldsymbol{V}_0, a_0, b_0)$, the posterior $p(\theta|\mathcal{D}_i)$ is the normal inverse-gamma distribution $\texttt{NIG}(\boldsymbol{w}_i, \boldsymbol{V}_i, a_0 + c_i/2, b_i)$ where $c_i$ is the number of data points and

$$
\boldsymbol{w}_i = \boldsymbol{V}_i \boldsymbol{X}_i^\top \boldsymbol{y}_i \qquad \boldsymbol{V}_i = \left(\boldsymbol{V}_0^{-1} + \boldsymbol{X}_i^\top \boldsymbol{X}_i\right)^{-1} \qquad b_i = b_0 + (1/2)\left[\boldsymbol{y}_i^\top \boldsymbol{y}_i - \boldsymbol{w}_i^\top \boldsymbol{V}_i^{-1} \boldsymbol{w}_i\right]
$$

can be computed directly from $\boldsymbol{s}_i$. The posterior belief $p(\theta|\mathcal{D}_i, \mathcal{D}_j)$ given parties $i$ and $j$'s dataset can be similarly computed using the SS of their pooled dataset, $\boldsymbol{s}_{ij}$. As the SS $\boldsymbol{s}_{ij}$ works out to $\boldsymbol{s}_i + \boldsymbol{s}_j$, we only need $\boldsymbol{s}_i$ and $\boldsymbol{s}_j$ from party $i$ and $j$ instead of their private datasets.

**Generalized Linear Model (GLM).**  A *generalized linear model* (GLM) generalizes a linear model by introducing an inverse link function $\Upsilon$. The probability of observing the output $y$ given input $\boldsymbol{x} = (x_{(1)}, \ldots, x_{(w)})$ and model weights $\theta$ depends on their dot product

$$
p(y|\boldsymbol{x}, \theta) = p(y|\Upsilon(\boldsymbol{x}^\top \theta)) .
$$

Next, we define the *GLM mapping function* $\upsilon$ to the log-likelihood of observing $y$ given the GLM model. Formally,

$$
\upsilon(y, \boldsymbol{x}^\top \theta) \triangleq \log p(y|\Upsilon(\boldsymbol{x}^\top \theta)) .
$$

As an example, logistic regression is a GLM with $\Upsilon$ defined as the sigmoid function and $p(y = \pm 1|\texttt{sigmoid}(\boldsymbol{x}^\top \theta))$ follows a Bernoulli distribution. As the non-linearity of $\Upsilon$ disrupts the exponential family structure, logistic regression and other GLMs do not have sufficient statistics. Logistic regression's GLM mapping function $\upsilon_{\texttt{log}}(y, \boldsymbol{x}^\top \theta) = -\log(1 + \exp(-y\boldsymbol{x}^\top \theta))$.

[21] propose to approximate the *GLM mapping function* $\upsilon$ with an $M$-degree polynomial approximation $\upsilon_M$. $\upsilon_M$ is an exponential family model with sufficient statistics $g(d) = \left\{\prod_{i=1}^{w}(yx_{(i)})^{m_i} | \sum_i m_i \leq M, \forall i \; m_i \in \mathbb{Z}_0^+\right\}$. These SS are the *polynomial approximate sufficient statistics* for GLMs. For example, when $M = 2$ and $\boldsymbol{x} = (x_{(1)}, x_{(2)})$, $g(d) = \left[1, x_{(1)}y, x_{(2)}y, x_{(1)}^2 y^2, x_{(2)}^2 y^2, x_{(1)}x_{(2)}y^2\right]$.

## A.2 Differential Privacy

*Remark 1.* Our work aims to ensure *example-level DP* for each collaborating party: A party updating/adding/deleting a single datum will only change the perturbed SS visible to the mediator and the corresponding belief of the model parameters in a provably minimal way. We are not ensuring *user-level DP*: The belief of model parameters only changes minimally after removing a collaborating party's (or a user/data owner's) dataset, possibly with multiple data points [35].

Intuitively, a DP algorithm $\mathcal{R} : \mathcal{D} \rightarrow o$ guarantees that each output $o$ is almost equally likely regardless of the inclusion or exclusion of a data point $d$ in $\mathcal{D}$. This will allay privacy concerns and incentivize a data owner to contribute its data point $d$ since even a knowledgeable attacker cannot infer the presence or absence of $d$.

The works on noise-aware inference [4, 27] assume that the input $x$ and output $y$ of any data point have known *bounded* ranges. We will start by introducing our domain-dependent definitions:

**Definition A.1** (Neighboring datasets). Two datasets $\mathcal{D}$ and $\mathcal{D}'$ are neighboring if $\mathcal{D}'$ can be obtained from $\mathcal{D}$ by replacing a single data point. The total number of data points and all other data points are the same.

**Definition A.2** (Sensitivity [11]). The sensitivity of a function $g$ that takes in dataset $\mathcal{D}_k$ quantifies the maximum impact a data point can have on the function output. The $\ell_1$-sensitivity $\Delta_1(g)$ and $\ell_2$-sensitivity $\Delta_2(g)$ measure the impact using the $\ell_1$ and $\ell_2$ norm, respectively. Given that $\mathcal{D}'_i$ must be a neighboring dataset of $\mathcal{D}_i$,

$$\Delta_1(g) \triangleq \max_{\mathcal{D}_i, \mathcal{D}'_i} \|g(\mathcal{D}_i) - g(\mathcal{D}'_i)\|_1 \ ,$$

$$\Delta_2(g) \triangleq \max_{\mathcal{D}_i, \mathcal{D}'_i} \|g(\mathcal{D}_i) - g(\mathcal{D}'_i)\|_2 \ .$$

In our problem, $g$ computes the exact SS $s_i$ for $\mathcal{D}_i$. The sensitivity can be known/computed if the dataset is normalized and the feature ranges are bounded.

We start with the definition of $\epsilon$-differential privacy. The parameter $\epsilon$ bounds how much privacy is lost by releasing the algorithm's output.

**Definition A.3** (Pure $\epsilon$-DP [11]). A randomized algorithm $\mathcal{R} : \mathcal{D} \rightarrow o$ with range $\mathcal{O}$ is $\epsilon$-DP if for all neighboring datasets $\mathcal{D}$ and $\mathcal{D}'$ and possible output subset $\mathcal{O} \subset Range(\mathcal{R})$,

The Laplace mechanism [11] is an $\epsilon$-DP algorithm. Instead of releasing the exact SS $s_i$, the mechanism will output a sample of the perturbed SS $o_i \sim \text{Laplace}(s_i, (\Delta_1(g)/\epsilon) \, I)$.

A common relaxation of $\epsilon$-differential privacy is $(\epsilon, \delta)$-differential privacy. It can be interpreted as $\epsilon$-DP but with a failure of probability at most $\delta$.

**Definition A.4** ($(\epsilon, \delta)$-DP). A randomized algorithm $\mathcal{R} : \mathcal{D} \rightarrow o$ with range $\mathcal{O}$ is $(\epsilon, \delta)$-differentially private if for all neighboring datasets $\mathcal{D}$ and $\mathcal{D}'$ and possible output subset $\mathcal{O} \subset Range(\mathcal{R})$,

$$P(\mathcal{R}(\mathcal{D}) \in \mathcal{O}) \leq e^\epsilon P(\mathcal{R}(\mathcal{D}') \in \mathcal{O}) + \delta \ .$$

The Gaussian mechanism is an $(\epsilon, \delta)$-DP algorithm. The variance of the Gaussian noise to be added can be computed by the analytic Gaussian mechanism algorithm [2].

In the main paper, we have also discussed another relaxation of $\epsilon$-differential privacy that is reproduced below:

**Definition A.5** (Rényi DP [38]). A randomized algorithm $\mathcal{R} : \mathcal{D} \rightarrow o$ is $(\lambda, \epsilon)$-Rényi differentially private if for all neighboring datasets $\mathcal{D}$ and $\mathcal{D}'$, the Rényi divergence of order $\lambda > 1$ is $D_\lambda(\mathcal{R}(\mathcal{D}) \,\|\, \mathcal{R}(\mathcal{D}')) \leq \epsilon$ where

$$D_\lambda(\mathcal{R}(\mathcal{D}) \,\|\, \mathcal{R}(\mathcal{D}')) \triangleq \frac{\log \mathbb{E}_{o \sim \mathcal{R}(\mathcal{D}')} \left[ \frac{P(\mathcal{R}(\mathcal{D}) = o)}{P(\mathcal{R}(\mathcal{D}') = o)} \right]^\lambda}{\lambda - 1} \ .$$

When $\lambda = \infty$, Rényi DP becomes pure $\epsilon$-DP. Decreasing $\lambda$ emphasizes less on unlikely large values and emphasizes more on the average value of the privacy loss random variable $\log \left[ P(\mathcal{R}(\mathcal{D}) = \boldsymbol{o}) / P(\mathcal{R}(\mathcal{D}') = \boldsymbol{o}) \right]$ with $\boldsymbol{o} \sim \mathcal{R}(\mathcal{D}')$.

The Gaussian mechanism is a $(\lambda, \epsilon)$-Rényi DP algorithm. Instead of releasing the exact SS $\boldsymbol{s}_i$, the mechanism will output a sample of the perturbed SS $\boldsymbol{o}_i \sim \mathcal{N} \left( \boldsymbol{s}_i, \ 0.5 \, (\lambda/\epsilon) \, \Delta_2^2(g) \, \boldsymbol{I} \right)$.

**Post-processing.** A common and important property of all DP algorithms/mechanisms is their *robustness to post-processing*: Processing the output of a DP algorithm $\mathcal{R}$ without access to the underlying dataset will retain the same privacy loss and guarantees [12].

**Choosing Rényi-DP over $(\epsilon, \delta)$-DP.** In our work, we consistently use the Gaussian mechanism in all the experiments, like in that of [27]. We choose Rényi DP over $(\epsilon, \delta)$-DP due to the advantages stated below:

- Rényi-DP is a stronger DP notion according to [38]: While $(\epsilon, \delta)$-DP allows for a complete failure of privacy guarantee with probability of at most $\delta$, Rényi-DP does not and the privacy bound is only loosened more for less likely outcomes. Additionally, [38] claims that it is harder to analyze and optimize $(\epsilon, \delta)$-DP due to the trade-off between $\epsilon$ and $\delta$. More details can be found in [38].
- Rényi-DP supports easier composition: In a collaborative ML framework, each party $i$ may need to release multiple outputs on the same dataset $\mathcal{D}_i$ such as the SS and other information for preprocessing steps (e.g., principal component analysis). Composition rules bound the total privacy cost $\hat{\epsilon}$ of releasing multiple outputs of differentially private mechanisms. It is harder to keep track of the total privacy cost when using $(\epsilon, \delta)$-DP due to advanced composition rules and the need to choose from a wide selection of possible $(\epsilon(\delta), \delta)$ [38]. In contrast, the composition rule (i.e., Proposition 1 in [38]) is straightforward: When $\lambda$ is a constant, the $\epsilon$ of different mechanisms can simply be summed.

Note that the contribution of our work will still hold for $(\epsilon, \delta)$-DP (using the Gaussian mechanism) and $\epsilon$-DP (using the Laplace mechanism) with some modifications of the inference process and proofs.

*Remark 2.* Our work is in the same spirit as local DP (and we also think that no mediator can be trusted to directly access any party's private dataset) but does not strictly satisfy the definition of local DP (see Def. A.6). In the definition, the local DP algorithm takes in a single input/datum and ensures the privacy of its output — the perturbation mechanism is applied to every input independently. In contrast, in our case, a party may have multiple inputs and the perturbation mechanism is only applied to their aggregate statistics. Thus, a datum owner (e.g., a patient of a collaborating hospital) enjoys weaker privacy in our setting than the local DP setting.

**Definition A.6** ($\epsilon$-Local DP [61])**.** A randomized algorithm $\mathcal{R}$ is $\epsilon$-local DP if for any pair of data points $d, d' \in \mathcal{D}$ and for any possible output $\mathcal{O} \subset Range(\mathcal{R})$,

$$P(\mathcal{R}(d) \in \mathcal{O}) \leq e^\epsilon P(\mathcal{R}(d') \in \mathcal{O}) \ .$$

### A.3 DP Noise-Aware Inference

DP mechanisms introduce randomness and noise to protect the output of a function. *Noise-naive* techniques ignore the added noise in downstream analysis. In contrast, *noise-aware* techniques account for the noise added by the DP mechanism.

Consider a probabilistic model where the model parameters $\theta$ generate the dataset $\mathcal{D}_i$ which then generates the exact and perturbed *sufficient statistics* of each party $i$, which are modeled as random variables $S_i$ and $O_i$, respectively. The exact SS $\boldsymbol{s}_i$ and perturbed SS $\boldsymbol{o}_i$ computed by party $i$ are *realizations* of $S_i$ and $O_i$, respectively. As the mediator cannot observe $i$'s exact SS, $S_i$ is a *latent* random variable. Instead, the mediator observes $i$'s perturbed SS $O_i$ which also contains noise $Z_i$ added by the DP mechanism, i.e., $O_i \triangleq S_i + Z_i$. The Gaussian mechanism to ensure $(\lambda, \epsilon)$-Rényi DP sets $Z_i \sim \mathcal{N} \left( \boldsymbol{0}, \ 0.5 \, (\lambda/\epsilon) \, \Delta_2^2(g) \, \boldsymbol{I} \right)$. We depict the graphical model of our multi-party setting in Fig. 4.

**Differences between exact, noise-naive and noise-aware inference.** When the mediator observes the exact SS $\boldsymbol{s}_i$ from party $i$, the exact posterior belief $p(\theta | S_i = \boldsymbol{s}_i)$ can be computed in closed form based on App. A.1. However, when the mediator only observes the perturbed SS $\boldsymbol{o}_i$, the mediator can

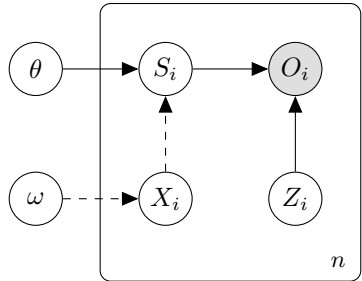

Figure 4: In the graphical model above, all parties share the same prior belief $p(\theta)$ of model parameters $\theta$ and prior belief $p(\omega)$ of data parameters $\omega$. The mediator models its beliefs of the SS of each party separately and only observes the perturbed SS $\boldsymbol{o}_i$ of every party $i \in N$ (thus, only $O_i$ is shaded). The sufficient statistic $S_i$ is generated from the model inputs $\boldsymbol{X}_i$ and the model output $\boldsymbol{y}_i$ (which depends on the model parameters $\theta$). We illustrate the relationship between $\omega$, $X_i$, and $S_i$ as dashed lines as they may be modeled differently in the various DP noise-aware inference methods. See [4, 27] for their respective graphical models and details.

only compute the noise-naive and noise-aware posterior beliefs instead. The *noise-naive* posterior belief $p(\theta|S_i = \boldsymbol{o}_i)$ will neither reflect the unobservability of the exact SS random variable $S_i$ accurately nor quantify the impact of the DP noise $Z_i$. In contrast, for any party, the *DP noise-aware posterior belief* $p(\theta|O_i = \boldsymbol{o}_i)$, conveniently abbreviated as $p(\theta|\boldsymbol{o}_i)$, will quantify the impact of the noise added by the DP mechanism. (For a coalition $C$ of parties, the DP noise-aware posterior belief is $p(\theta|\boldsymbol{o}_C) \triangleq p(\theta|\{O_i = \boldsymbol{o}_i\}_{i \in C})$, as described in Footnote 6.) The works of [3, 4, 27] have shown that DP noise-aware inference leads to a posterior belief that is better *calibrated* (i.e., lower bias and better quantification of uncertainty without overconfidence) and of higher *utility* (i.e., closer to the non-private posterior belief), thus a better predictive performance.

The main challenge of noise-aware inference lies in tractably approximating the integral $p(O_i = \boldsymbol{o}_i|\theta) = \int p(\boldsymbol{s}_i|\theta)p(\boldsymbol{o}_i|\boldsymbol{s}_i)\,\mathrm{d}\boldsymbol{s}_i$ and $p(\boldsymbol{s}_i|\theta) = \int_{\{\mathcal{D}=(\boldsymbol{X},\boldsymbol{y}):g(\mathcal{D})=\boldsymbol{s}_i\}} p(\boldsymbol{X},\boldsymbol{y}|\theta)\,\mathrm{d}\boldsymbol{X}\,\mathrm{d}\boldsymbol{y}$ over all datasets. [3, 4, 27] exploit the observation that as the SS sum $c$ individuals, the central limit theorem guarantees that the distribution $p(\boldsymbol{s}_i|\theta)$ can be well-approximated by the Gaussian distribution $\mathcal{N}(c\mu_g, c\Sigma_g)$ for large $c$ [4]. Here, $\mu_g$ and $\Sigma_g$ are the mean and covariance of an individual's SS. [3, 4, 27, 26] prescribe how to compute $\mu_g$ and $\Sigma_g$ in closed form from the sampled $\theta$ parameters and moments of $\boldsymbol{x}$ and set the noise of the DP mechanism based on a sensitivity analysis. To approximate the posterior $p(\theta|\boldsymbol{o})$, *Markov Chain Monte Carlo* (MCMC) sampling steps are needed. [4] propose to use Gibbs sampling, an algorithm that updates a group of parameters at a time and exploits conditional independence, for Bayesian linear regression models. [27] use the No-U-Turn [19] sampler and utilizes Hamiltonian dynamics to explore the parameter space more efficiently for generalized linear models. We describe the BLR Gibbs sampler adapted for our multi-party setting in Algo 1.

**Algorithm 1** BLR Gibbs sampler [4] from noise-aware posterior $p(\theta|O_N = \boldsymbol{o}_N) \propto \int \prod_{i \in N} [p(\boldsymbol{o}_i|\boldsymbol{s}_i)\, p(\boldsymbol{s}_i|\theta)]\, p(\theta)\, \mathrm{d}\boldsymbol{s}_1 \cdots \mathrm{d}\boldsymbol{s}_n$. The algorithm (repeatedly) sample the latent variables $S_i$, $\omega$ and $\theta$ sequentially.

---

**Require:** Shared prior $p(\theta)$ of model parameters, prior $p(\omega)$ of data parameters, data quantity $c_i$, shared perturbed SS realization $\boldsymbol{o}_i$, the Gaussian noise distribution of $Z_i$ for every party $i \in N$, number $b$ of burn-in samples, number $m$ of samples, Boolean parameter (shared) controlling if $p(\boldsymbol{x})$ is the same across parties.
  1: Sample the initial model parameters $\theta^{(0)}$ from the prior $p(\theta)$.
  2: Sample the data prior parameters $\omega^{(0)}$ from the prior $p(\omega)$.
  3: Compute the moments of $X_i$ based on $\omega$.
  4: **for** $t = 1, \ldots, b + m$ **do**
  5:     **for** $i = 1, \ldots, n$ **do**
  6:         Compute the normal approximation of $p(S_i|\theta)$, denoted as $p_{\mathcal{N}}(S_i|\theta)$, using the moments of $X_i$.
  7:         Sample $\boldsymbol{s}_i^{(t)}$ from the product of two multivariate Gaussians $p_{\mathcal{N}}(S_i|\theta)\, p(\boldsymbol{o}_i|S_i)$, which is also multivariate Gaussian.
  8:         **if not** shared **then**
  9:             Use information from $\boldsymbol{s}_i^{(t)}$ and $c_i$ to perform conjugate update on $p(\omega_i)$ to obtain $p(\omega_i|(\boldsymbol{s}_i^{(t)}, c_i))$. Sample $\omega_i^{(t)}$ and compute the moments of $X_i$.
 10:         **end if**
 11:     **end for**
 12:     **if** shared **then**
 13:         Use information from $(\boldsymbol{s}_i^{(t)}, c_i)_{i \in N}$ to perform conjugate update on $p(\omega)$ to obtain $p(\omega|(\boldsymbol{s}_i^{(t)}, c_i)_{i \in N})$. Sample $\omega^{(t)}$ and compute the moments of $X_i$.
 14:     **end if**
 15:     Use $(\boldsymbol{s}_i^{(t)}, c_i)_{i \in N}$ to perform conjugate update on $p(\theta)$ to obtain $p(\theta|(\boldsymbol{s}_i^{(t)}, c_i)_{i \in N})$.
 16:     Sample $\theta^{(t)}$ from $p(\theta|(\boldsymbol{s}_i^{(t)}, c_i)_{i \in N})$.
 17:     **if** $t > b$ **then**
 18:         Append $\theta^{(t)}$ to $\Theta$.
 19:     **end if**
 20: **end for**
 21: **return** $\Theta$

## B   Key Differences with Existing Data Valuation, Collaborative ML, and DP/FL Works

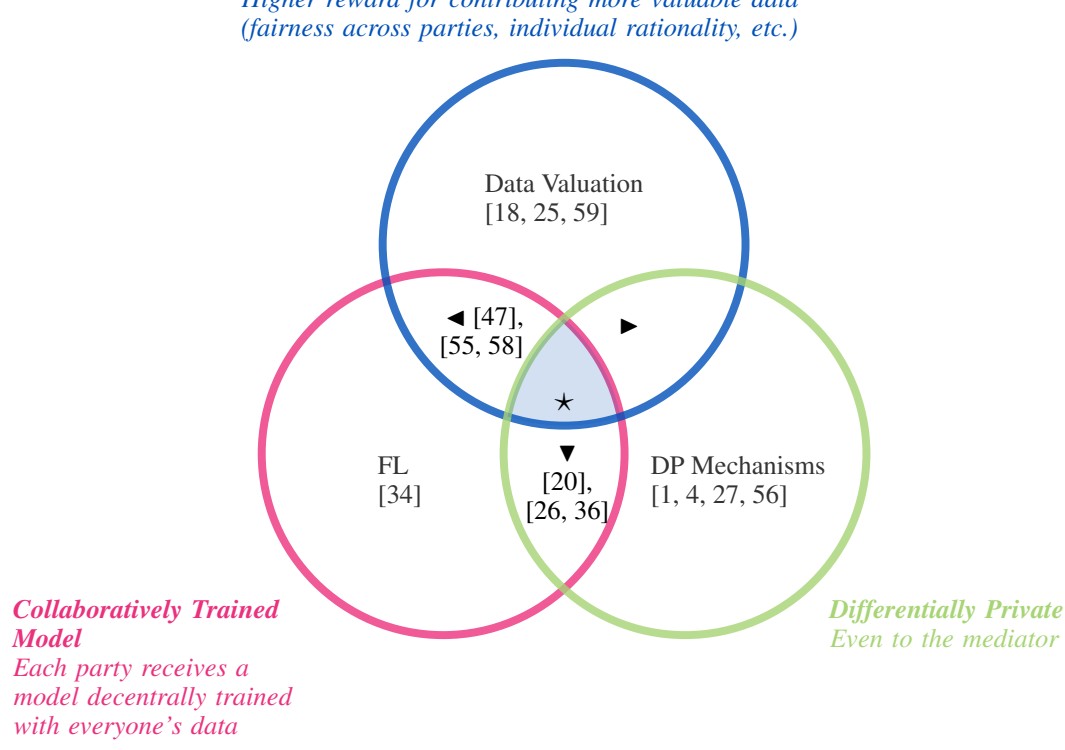

Figure 5: Our work, ⋆, uniquely satisfies all 3 desiderata. When parties share information computed from their data, we ensure that every party has at least its required DP w.r.t. the mediator, receives a collaboratively trained model, and receives a higher reward for sharing higher-quality data than the others.

It is not trivial to (i) add DP to ◄ while simultaneously enforcing a privacy-valuation trade-off, (ii) add data sharing incentives to ▼ (i.e., design valuation functions and rewards), and (iii) achieve ► as access to a party's dataset (or a coalition's datasets) is still needed for its valuation in [57].

*Difference with existing data valuation and collaborative ML works considering incentives.* Our work aims to additionally (A) offer parties assurance about privacy but (B) deter them from selecting excessive privacy guarantees. We achieve (A) by ensuring differential privacy (see definitions in App. A.2) through only collecting the noisier/perturbed version of each party's sufficient statistics (see App. A.1). To achieve (B), we must assign a lower valuation (and reward) to a noisier SS. Our insight is to combine noise-aware inference (that computes the posterior belief of the model parameters given the perturbed SS) with the Bayesian surprise valuation function. Lastly, (C) we propose a mechanism to generate model rewards (i.e., posterior samples of the model parameters) that attain the target reward value and are similar to the grand coalition's model.

*Difference with federated learning and differential privacy works.* Existing FL works have covered learning from decentralized data with DP guarantees. However, these works may not address the question: Would parties want to share their data? How do we get parties to share more to maximize the gain from the collaboration? Our work aims to address these questions and **incentivize** (A) parties to share more, higher-quality data and (B) select a weaker DP guarantee. To achieve (A), it is standard in data valuation methods [18, 25, 46] to use the Shapley value to value a party *relative* to the data of others as it considers a party's marginal contribution to all coalitions (subsets) of parties. This would require us to construct and value a trained model for each coalition $C \subseteq N$: To ease aggregation (and to avoid requesting more information or incurring privacy costs per coalition), we consider

sufficient statistics (see App. A.1). To achieve (B), we want a valuation function that provably ensures a lower valuation for a stronger DP guarantee. Our insight is to combine noise-aware inference (that computes the posterior belief of the model parameters given perturbed SS) with the Bayesian surprise valuation function. Lastly, like the works of [47, 51], (C) we generate a model reward that attains a target reward value (which parties can use for future predictions). Our model reward is in the form of posterior samples of the model parameters instead. We propose a new mechanism to control/generate model rewards that work using SS and preserve similarity to the grand coalition's model.

Fig. 5 shows how our work in this paper fills the gap in the existing works.

## C  Characteristic/Valuation Function

### C.1  Proofs of properties for valuation function

In this section, we will use the random variable notations defined in App. A. Moreover, we abbreviate the set of perturbed SS random variables corresponding to a coalition $C$ of parties as $O_C \triangleq \{O_i\}_{i \in C}$.

Let $\mathbb{H}(a)$ denote the entropy of the variable $a$.

**Relationship between KL divergence and information gain.**

$$\mathbb{I}(\theta; O_C) = \mathbb{E}_{\boldsymbol{o}_C \sim O_C}[D_{\mathrm{KL}}(p(\theta|\boldsymbol{o}_C); p(\theta))]$$
$$= \mathbb{H}(\theta) - \mathbb{E}_{\boldsymbol{o}_C \sim O_C}[\mathbb{H}(\theta|O_C = \boldsymbol{o}_C)] \ .$$

**Party monotonicity (V2).**  Consider two coalitions $C \subset C' \subseteq N$. By taking an expectation w.r.t. random vector $O_{C'}$,

$$\mathbb{E}_{\boldsymbol{o}_{C'} \sim O_{C'}}[v_C] = \mathbb{E}_{\boldsymbol{o}_C \sim O_C}[D_{\mathrm{KL}}(p(\theta|\boldsymbol{o}_C); p(\theta))] = \mathbb{I}(\theta; O_C) = \mathbb{H}(\theta) - \mathbb{H}(\theta|O_C)$$

and

$$\mathbb{E}_{\boldsymbol{o}_{C'} \sim O_{C'}}[v_{C'}] = \mathbb{E}_{\boldsymbol{o}_{C'} \sim O_{C'}}[D_{\mathrm{KL}}(p(\theta|\boldsymbol{o}_{C'}); p(\theta))] = \mathbb{I}(\theta; O_{C'}) = \mathbb{H}(\theta) - \mathbb{H}(\theta|O_C, O_{C' \setminus C}) \ .$$

Then, $\mathbb{E}_{\boldsymbol{o}_{C'} \sim O_{C'}}[v_{C'}] > \mathbb{E}_{\boldsymbol{o}_{C'} \sim O_{C'}}[v_C]$ as conditioning additionally on $O_{C' \setminus C}$ should not increase the entropy (i.e., $\mathbb{H}(\theta|O_C, O_{C' \setminus C}) \leq \mathbb{H}(\theta|O_C)$) due to the "information never hurts" bound for entropy [10].

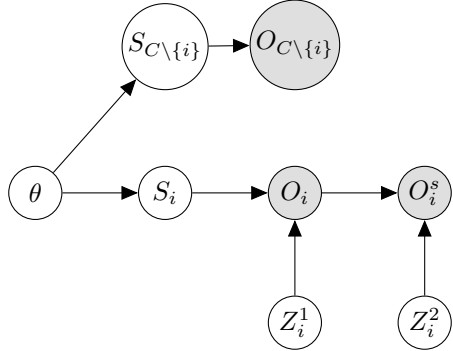

Figure 6: Graphical model to illustrate privacy-valuation trade-off (V3) where $O_i \triangleq S_i + Z_i^1$ and $O_i^s \triangleq O_i + Z_i^2$.

**Privacy-valuation trade-off (V3).**  Let $\epsilon_i^s < \epsilon_i$, and $Z_i^1$ and $Z_i^2$ be independent Gaussian distributions with mean 0 and, respectively, variance $a_i/\epsilon_i$ and $(a_i/\epsilon_i^s) - (a_i/\epsilon_i) > 0$ where $a_i \triangleq 0.5\,\lambda\,\Delta_2^2(g)$, function $g$ computes the exact SS $\boldsymbol{s}_i$ from local dataset $\mathcal{D}_i$, and $\Delta_2(g)$ denotes its $\ell_2$-sensitivity. Adding $Z_i^1$ to $S_i$ will ensure $(\lambda, \epsilon_i)$-DP while adding both $Z_i^1$ and *independent* $Z_i^2$ to $S_i$ is equivalent to adding Gaussian noise of variance $a_i/\epsilon_i^s$ to ensure $(\lambda, \epsilon_i^s)$-DP.[18] From the graphical model

---

[18]Adding or subtracting *independent* noise will lead to a random variable with a *higher* variance. Thus, we cannot model the random variable $O_i$ of a lower variance $a_i/\epsilon_i$ to ensure $(\lambda, \epsilon_i)$-DP as $O_i^s - Z_i^2$.

in Fig. 6 and the Markov chain $\theta \to O_i \to O_i^s$, the following conditional independence can be observed: $\theta \perp\!\!\!\perp O_i^s \mid O_i$. By the *data processing inequality*, no further processing of $O_i$, such as the addition of noise, can increase the information of $\theta$. Formally, $\mathbb{I}(\theta; O_i) \geq \mathbb{I}(\theta; O_i^s)$. Simultaneously, $\theta \not\perp\!\!\!\perp O_i \mid O_i^s$. Hence, $\mathbb{I}(\theta; O_i) \neq \mathbb{I}(\theta; O_i^s) \Rightarrow (\mathbb{I}(\theta; O_i) > \mathbb{I}(\theta; O_i^s))$.

To extend to any coalition $C$ containing $i$, by the chain rule of mutual information,

$$\mathbb{I}\big(\theta; O_i, O_i^s, O_{C\setminus\{i\}}\big) = \mathbb{I}\big(\theta; O_i, O_{C\setminus\{i\}}\big) + \mathbb{I}\big(\theta; O_i^s | O_i, O_{C\setminus\{i\}}\big)$$
$$= \mathbb{I}\big(\theta; O_i^s, O_{C\setminus\{i\}}\big) + \mathbb{I}\big(\theta; O_i | O_i^s, O_{C\setminus\{i\}}\big) \ .$$

As conditional independence $\theta \perp\!\!\!\perp O_i^s \mid O_i, O_{C\setminus\{i\}}$ and dependence $\theta \not\perp\!\!\!\perp O_i \mid O_i^s, O_{C\setminus\{i\}}$ still hold, $\mathbb{I}\big(\theta; O_i^s | O_i, O_{C\setminus\{i\}}\big) = 0$ and $\mathbb{I}\big(\theta; O_i | O_i^s, O_{C\setminus\{i\}}\big) > 0$, respectively. It follows from the above expression that $\mathbb{I}(\theta; O_C) > \mathbb{I}(\theta; O_i^s, O_{C\setminus\{i\}})$, which implies $\mathbb{E}_{O_C}[v_C] > \mathbb{E}_{O_{C\setminus\{i\}}, O_i^s}[v_C^s]$ . For future work, the proof can be extended to other DP mechanisms.

## C.2 Proof of Remark in Sec. 3

Let the alternative valuation of a coalition $C$ be $v_C' \triangleq D_{\mathrm{KL}}(p(\theta|\boldsymbol{o}_N); p(\theta)) - D_{\mathrm{KL}}(p(\theta|\boldsymbol{o}_N); p(\theta|\boldsymbol{o}_C))$. Then, $v_\emptyset' = 0$ and $v_N' = D_{\mathrm{KL}}(p(\theta|\boldsymbol{o}_N); p(\theta))$. It can be observed that

- Unlike $v_C$, $v_C'$ may be negative.
- Unlike $v_N$, $v_N'$ is guaranteed to have the highest valuation as the minimum KL divergence $D_{\mathrm{KL}}(p(\theta|\boldsymbol{o}_N); q(\theta))$ is 0 only when $q(\theta) = p(\theta|\boldsymbol{o}_N)$. This is desirable when we want the grand coalition to be more valuable than the other coalitions but odd when we consider the non-private posterior $q(\theta) = p(\theta|\boldsymbol{s}_N)$: Intuitively, the model computed using $\boldsymbol{s}_N$ should be more valuable using $v'$ than that computed using the perturbed SS $\boldsymbol{o}_N$.

By taking an expectation w.r.t. $\boldsymbol{o}_N$,

$$\mathbb{E}_{p(O_N)}[v_C'] = \mathbb{I}(\theta; O_N) - \mathbb{E}_{\boldsymbol{o}_C \sim p(O_C)}\Big[\mathbb{E}_{\boldsymbol{o}_{N\setminus C} \sim p(O_{N\setminus C}|\boldsymbol{o}_C)}\big[D_{\mathrm{KL}}\big(p(\theta|\boldsymbol{o}_N = \{\boldsymbol{o}_{N\setminus C}, \boldsymbol{o}_C\}); p(\theta|\boldsymbol{o}_C)\big)\big]\Big]$$

$$= \mathbb{I}(\theta; O_N) - \mathbb{E}_{\boldsymbol{o}_C \sim p(O_C)}\Big[\mathbb{E}_{\boldsymbol{o}_{N\setminus C} \sim p(O_{N\setminus C}|\boldsymbol{o}_C)}\Big[\mathbb{E}_{\theta \sim p(\theta|\boldsymbol{o}_{N\setminus C}, \boldsymbol{o}_C)}\Big[\log \frac{p(\theta|\boldsymbol{o}_{N\setminus C}, \boldsymbol{o}_C)}{p(\theta|\boldsymbol{o}_C)}\Big]\Big]\Big]$$

$$\overset{(i)}{=} \mathbb{I}(\theta; O_N) - \mathbb{E}_{\boldsymbol{o}_C \sim p(O_C)}\Big[\mathbb{E}_{\theta, \boldsymbol{o}_{N\setminus C} \sim p(\theta, O_{N\setminus C}|\boldsymbol{o}_C)}\Big[\log \frac{p(\theta, \boldsymbol{o}_{N\setminus C}|\boldsymbol{o}_C)}{p(\theta|\boldsymbol{o}_C) \, p(\boldsymbol{o}_{N\setminus C}|\boldsymbol{o}_C)}\Big]\Big]$$

$$= \mathbb{I}(\theta; O_N) - \mathbb{E}_{\boldsymbol{o}_C \sim p(O_C)}\big[D_{\mathrm{KL}}\big(p(\theta, O_{N\setminus C}|\boldsymbol{o}_C); p(\theta|\boldsymbol{o}_C) \, p(O_{N\setminus C}|\boldsymbol{o}_C)\big)\big]$$

$$\overset{(ii)}{=} \mathbb{I}(\theta; O_N) - \mathbb{I}\big(\theta; O_{N\setminus C}|O_C\big)$$

$$= \mathbb{I}(\theta; O_C) = \mathbb{E}_{p(O_N)}[v_C] \ .$$

In equality (i) above, we multiply both the numerator and denominator within the log term by $p(\boldsymbol{o}_{N\setminus C}|\boldsymbol{o}_C)$ and consider the expectation of the joint distribution since by the chain rule of probability, $p(\theta, O_{N\setminus C}|\boldsymbol{o}_C) = p(O_{N\setminus C}|\boldsymbol{o}_C) \, p(\theta|O_{N\setminus C}, \boldsymbol{o}_C)$. Equality (ii) is due to the definition of conditional mutual information.

## C.3 KL Estimation of Valuation Function

KL estimation is only a tool and not the focus of our work. Our valuation will become more accurate and computationally efficient as KL estimation tools improve.

**Recommended - nearest-neighbors [45, 54].** Given $\Theta^{\mathrm{post}}$ and $\Theta^{\mathrm{prior}}$ which consists of $m$ samples of $\theta$ (with dimension $d$) from, respectively, the posterior $p(\theta|\boldsymbol{o}_C)$ and prior $p(\theta)$, we estimate the KL divergence as

$$\frac{d}{m} \sum_{\theta \in \Theta^{\mathrm{post}}} \log \frac{\delta_k^{\mathrm{prior}}(\theta)}{\delta_k^{\mathrm{post}}(\theta)} + \log \frac{m}{m-1}$$

where $\delta_k^{\mathrm{post}}(\theta)$ is the distance of the sampled $\theta$ to its $k$-th nearest neighbor in $\Theta^{\mathrm{post}}$ (excluding itself) and $\delta_k^{\mathrm{prior}}(\theta)$ is the distance of the sampled $\theta$ to the $k$-th nearest neighbor in $\Theta^{\mathrm{prior}}$.

The number $k$ of neighbors is tunable and analyzed in the follow-up work of [49]. As the number $m$ of samples increases, the bias and variance decrease. The convergence rate is analysed by [66]. Moreover, the estimate converges almost surely [45] and is consistent [54] for *independent and identically distributed (i.i.d.) samples*. Furthermore, as the KL divergence is invariant to metric reparameterizations, the bias can be reduced by changing the distance metric [42, 54].

To generate i.i.d. samples, we suggest the usage of the NUTS sampler or thinning (keeping only every $t$-th sample). We observe that if the samples from $\theta \mid \boldsymbol{o}_C$ are non-independent, i.e., correlated and close to the previous sample, we may underestimate its distance to the $k$-th distinct neighbor in $\theta \mid \boldsymbol{o}_C$, $\delta_k^{\text{post}}(\theta)$, and thus overestimate the KL divergence. This is empirically verified in Table 2. We have also observed that the KL divergence may be underestimated when the posterior is concentrated at a significantly different mean from the prior.

**Recommended for large $\epsilon$ - approximate $p(\theta|\boldsymbol{o}_C)$ using maximum likelihood distribution from the $p(\theta)$'s exponential family.** When a small noise is added to ensure weak DP, we can approximate $p(\theta|\boldsymbol{o}_C)$ with a distribution $q$ from the same exponential family as $p(\theta|\boldsymbol{s}_C)$. We can (i) determine $q$'s parameters via *maximum likelihood estimation* (MLE) from the Gibbs samples[19] and (ii) compute the KL divergence in closed form.

However, the KL estimate is inaccurate (i.e., large bias) when the distribution $q$ is a poor fit for the posterior $p(\theta|\boldsymbol{o}_C)$. Future work can consider using *normalizing flows* as $q$ to improve the fit, reduce the estimation bias, and work for a larger range of DP guarantees $\epsilon$. However, this KL estimation method may be computationally slow and risks overfitting.

**Probabilistic Classification.** Let the binary classifier $f : \Theta \to [0, 1]$ (e.g., a neural network) discriminate between samples from two densities $q_1(\theta)$ (here, the posterior $p(\theta|\boldsymbol{o}_C)$) and $q_0(\theta)$ (here, the prior $p(\theta)$) and output the probability that $\theta$ comes from $q_1(\theta)$. Concretely, we label the $m$ samples from $q_1(\theta)$ and $q_0(\theta)$ with $y = 1$ and $y = 0$, respectively. By Bayes' rule, the density ratio is

$$\frac{q_1(\theta)}{q_0(\theta)} = \frac{p(\theta|y=1)}{p(\theta|y=0)} = \frac{p(y=1|\theta)}{p(y=0|\theta)} = \frac{p(y=1|\theta)}{1-p(y=1|\theta)} \ .$$

Optimizing a *proper scoring rule* such as minimizing the binary cross-entropy loss should return the Bayes optimal classifier $f^*(\theta) = p(y = 1|\theta)$. The KL estimate is then computed as the mean log-density ratio over samples from $q_1(\theta)$. As the log-density ratio is $\texttt{sigmoid}^{-1}(p(y=1|\theta))$, when $f$ is a neural network with $\texttt{sigmoid}$ as the last activation layer, we can use the logits before activation directly.

However, with only limited finite samples $m$ and a large separation between the distributions $q_1$ and $q_0$, the density ratio and KL estimate may be highly inaccurate [8]: Intuitively, the finite samples may be linearly separable and the loss is minimized by setting the logits of samples from $q_1(\theta)$ (hence KL) to infinity (i.e., classify it overconfidently with probability 1). As the separation between the distributions $q_1$ and $q_0$ increases, exponentially more training samples may be needed to obtain samples between $q_1$ and $q_0$ [8]. Moreover, as training may not produce the Bayes optimal classifier, there is also an issue of larger variance across runs.

# D    Reward Scheme for Ensuring Incentives

## D.1    Incentives based on Coalition Structure

Instead of assuming the grand coalition $N$ form, we can consider the more general case where parties team up and partition themselves into a *coalition structure $CS$*. Formally, $CS$ is a set of coalitions such that $\bigcup_{C \in CS} C = N$ and $C \cap C' = \emptyset$ for any $C, C' \in CS$ and $C \neq C'$. The following incentives are modified below:

**P2** For any coalition $C \in CS$, there is a party $i \in C$ whose model reward is the coalition $C$'s posterior, i.e., $q_i(\theta) = p(\theta|\boldsymbol{o}_C)$. It follows that $r_i = v_C$ as in R2 of [47].

**P5** Among multiple model rewards $q_i(\theta)$ whose value $r_i$ equates the target reward $r_i^*$, we secondarily prefer one with a higher similarity $r'_{i,C} = -D_{KL}(p(\theta|\boldsymbol{o}_C); q_i(\theta))$ to the coalition's posterior $p(\theta|\boldsymbol{o}_C)$ where $i \in C$.

---

[19]The distribution $q$ from MLE minimizes the KL divergence $D_{\text{KL}}(p(\theta|\boldsymbol{o}_C); q(\theta))$.

## D.2 Fairness Axioms

The fairness axioms from the work of [47] are reproduced below:

F1 **Uselessness.** If including the data or sufficient statistic of party $i$ does not improve the quality of a model trained on the aggregated data of any coalition (e.g., when $\mathcal{D}_i = \emptyset$, $c_i = 0$), then party $i$ should receive a valueless model reward: For all $i \in N$,

$$(\forall C \subseteq N \setminus \{i\} \ \ v_{C \cup \{i\}} = v_C) \Rightarrow r_i = 0 .$$

F2 **Symmetry.** If including the data or sufficient statistic of party $i$ yields the same improvement as that of party $j$ in the quality of a model trained on the aggregated data of any coalition (e.g., when $\mathcal{D}_i = \mathcal{D}_j$), then they should receive equally valuable model rewards: For all $i, j \in N$ s.t. $i \neq j$,

$$(\forall C \subseteq N \setminus \{i,j\} \ \ v_{C \cup \{i\}} = v_{C \cup \{j\}}) \Rightarrow r_i = r_j .$$

F3 **Strict Desirability [33].** If the quality of a model trained on the aggregated data of at least a coalition improves more by including the data or sufficient statistic of party $i$ than that of party $j$, but the reverse is not true, then party $i$ should receive a more valuable model reward than party $j$: For all $i, j \in N$ s.t. $i \neq j$,

$$(\exists B \subseteq N \setminus \{i,j\} \ \ v_{B \cup \{i\}} > v_{B \cup \{j\}}) \wedge$$
$$(\forall C \subseteq N \setminus \{i,j\} \ \ v_{C \cup \{i\}} \geq v_{C \cup \{j\}}) \Rightarrow r_i > r_j .$$

F4 **Strict Monotonicity.** If the quality of a model trained on the aggregated data of at least a coalition containing party $i$ improves (e.g., by including more data of party $i$), *ceteris paribus*, then party $i$ should receive a more valuable model reward than before: Let $\{v_C\}_{C \in 2^N}$ and $\{\tilde{v}_C\}_{C \in 2^N}$ denote any two sets of values of data over all coalitions $C \subseteq N$, and $r_i$ and $\tilde{r}_i$ be the corresponding values of model rewards received by party $i$. For all $i \in N$,

$$(\exists B \subseteq N \setminus \{i\} \ \ \tilde{v}_{B \cup \{i\}} > v_{B \cup \{i\}}) \wedge$$
$$(\forall C \subseteq N \setminus \{i\} \ \ \tilde{v}_{C \cup \{i\}} \geq v_{C \cup \{i\}}) \wedge$$
$$(\forall A \subseteq N \setminus \{i\} \ \ \tilde{v}_A = v_A) \wedge (\tilde{v}_N > r_i) \Rightarrow \tilde{r}_i > r_i .$$

## D.3 Remark on Rationality

Let $v_{s_i}$ denote the Bayesian surprise party $i$'s *exact* SS $s_i$ elicits from the prior belief of model parameter. We define **Stronger Individual Rationality** (SIR, the strengthened version of P4) as: each party should receive a model reward that is more valuable than the model trained on its exact SS alone: $\forall i \in N r_i^* \geq v_{s_i}$.

We consider two potential solutions to achieve stronger individual rationality and explain how they fall short.

- Each party $i$ declares the value $v_{s_i}$ and the mediator selects a smaller $\rho$ to guarantee SIR. SIR is infeasible when the grand coalition's posterior is less valuable than party $i$'s model based on exact SS, i.e., $v_N < v_{s_2}$. In Fig. 2, we observe that when party 2 selects a small $\epsilon_2$, $v_N$ is less than $v_{s_2}$ which can approximated by $v_2$ under large $\epsilon_2$, i.e., the right end point of the blue line. Instead, SIR is only empirically achievable when party 2 and others select a weaker DP guarantee as in Fig. 10 in App. H.5. Note that our work only incentivizes weaker DP guarantees and does not restrict parties' choice of DP guarantees.
- The mediator should reward party $i$ with perturbed SS $t_j^i$ (for Sec. 5.1) or $\kappa_i o_j, \kappa_i c_j, \kappa_i Z_j$ (for Sec. 5.2) for every other party $j \neq i$. Party $i$ can then combine these with its exact $s_i$ to guarantee SIR.
  This approach achieves SIR at the expense of truthfulness. As party $i$ perturbed SS $o_i$ is not used to generate its own model reward, party $i$ may be less deterred (hence more inclined) to submit less informative or fake SS.

SIR is not needed when each party prefers training a DP model even when alone. SIR may be desired in other scenarios. However, our approach does not use alternative solutions to satisfy SIR as we prioritize incentivizing parties to (i) truthfully submit informative perturbed SS that they would use for future predictions, while (ii) not compromising for weak DP guarantees.

# E   Details on Reward Control Mechanisms

In the subsequent proofs, any likelihood $p^{\kappa_i}(\cdot)$ should be interpreted as $[p(\cdot)]^{\kappa_i}$: We only raise likelihoods (of data conditioned on model parameters) to the power of $\kappa_i$.

## E.1   Likelihood Tempering and Scaling SS Equivalence Proof

Let $g$ denote the function that maps any data point $d_l$ or dataset $\mathcal{D}_k$ to its sufficient statistic. For any data point $d_l$, we assume that the data likelihood $p(d_l|\theta)$ is from an exponential family with natural parameters $\theta$ and sufficient statistic $g(d_l)$. The data likelihood $p(d_l|\theta)$ can be expressed in its natural form:

$$p(d_l|\theta) = h(d_l) \exp\left[g(d_l) \cdot \theta - A(\theta)\right]$$

where $\boldsymbol{a} \cdot \boldsymbol{b} \triangleq \boldsymbol{a}^\top \boldsymbol{b}$ denotes the dot product between two vectors.

Next, we assume that $p(\theta)$ is the conjugate prior[20] for $p(\mathcal{D}_k|\theta)$ with natural parameters $\eta$ and the sufficient statistic mapping function $T : \theta \to \left[\theta^\top, -A(\theta)\right]^\top$. Then, for $c_k$ data points which are conditionally independent given the model parameters $\theta$,

$$p(\theta|\{d_l\}_{l=1}^{c_k}) \propto p(d_1|\theta)\dots p(d_{c_k}|\theta)\, p(\theta|\eta)$$

$$\propto \left[\left(\prod_{l=1}^{c_k} h(d_l)\right) \exp\left[\underbrace{\sum_{l=1}^{c_k} g(d_l)}_{g(\mathcal{D}_k)} \cdot \theta - c_k A(\theta)\right]\right] [h(\theta) \exp\left[T(\theta) \cdot \eta - B(\eta)\right]]$$

$$\propto \exp\left[g(\mathcal{D}_k) \cdot \theta - c_k A(\theta) + T(\theta) \cdot \eta - B(\eta)\right]$$

$$\propto \exp\left[\left(\left[g(\mathcal{D}_k)^\top, c_k\right]^\top + \eta\right) \cdot T(\theta) - C(\eta)\right]$$

where $C(\eta)$ is chosen such that the distribution is normalized.

Substituting the above SS formulae into (1), the normalized posterior distribution (after tempering the likelihood) is

$$q_i(\theta) \propto p^{\kappa_i}(d_1|\theta)\dots p^{\kappa_i}(d_{c_k}|\theta)\, p(\theta|\eta)$$

$$\propto \left[\left(\prod_{l=1}^{c_k} h(d_l)\right)^{\kappa_i} \exp\left[\kappa_i\left[g(\mathcal{D}_k) \cdot \theta - c_k A(\theta)\right]\right]\right] [h(\theta) \exp\left[T(\theta) \cdot \eta - B(\eta)\right]]$$

$$\propto \exp\left[\kappa_i g(\mathcal{D}_k) \cdot \theta - \kappa_i c_k A(\theta) + T(\theta) \cdot \eta - B(\eta)\right]$$

$$\propto \exp\left[\left(\left[\kappa_i g(\mathcal{D}_k)^\top, \kappa_i c_k\right]^\top + \eta\right) \cdot T(\theta) - C'(\eta)\right]$$

where $C'(\eta)$ is chosen such that the distribution is normalized.

Thus, tempering the likelihood by $\kappa_i$ is equivalent to scaling the SS $g(\mathcal{D}_k)$ and data quantity $c_k$. We additionally proved that the normalized posterior can be obtained from the scaled SS and data quantity via the conjugate update.

**Bayesian Linear Regression (BLR).**   The Bayesian linear regression model was introduced in App. A.1 To recap, BLR model parameters $\theta$ consists of the weight parameters $\boldsymbol{w} \in \mathbb{R}^w$ and the noise variance $\sigma^2$. BLR models the relationship between the concatenated output vector $\boldsymbol{y}$ and the design matrix $\boldsymbol{X}$ as $\boldsymbol{y} = \boldsymbol{X}\boldsymbol{w} + \mathcal{N}(0, \sigma^2\boldsymbol{I})$. The tempered likelihood

$$p^{\kappa_i}(\boldsymbol{y}|\boldsymbol{X}, \boldsymbol{w}, \sigma^2) \propto \left[(2\pi\sigma^2)^{-\frac{c}{2}} \exp\left(-\frac{(\boldsymbol{y} - \boldsymbol{X}\boldsymbol{w})^\top(\boldsymbol{y} - \boldsymbol{X}\boldsymbol{w})}{2\sigma^2}\right)\right]^{\kappa_i}$$

$$= (2\pi\sigma^2)^{-\frac{c\kappa_i}{2}} \exp\left[\frac{-1}{2\sigma^2}\kappa_i \boldsymbol{y}^\top\boldsymbol{y} + \frac{1}{\sigma^2}\boldsymbol{w}^\top \kappa_i \boldsymbol{X}^\top\boldsymbol{y} - \frac{1}{2\sigma^2}\boldsymbol{w}^\top \kappa_i \boldsymbol{X}^\top\boldsymbol{X}\boldsymbol{w}\right].$$

only depends on the scaled sufficient statistics $\kappa_i(\boldsymbol{y}^\top\boldsymbol{y}, \boldsymbol{X}^\top\boldsymbol{y}, \boldsymbol{X}^\top\boldsymbol{X})$. When the prior $p(\boldsymbol{\theta})$ follows the conjugate normal inverse-gamma distribution, the power posterior can be obtained from the scaled SS and data quantity via the conjugate update.

---

[20] $p(\theta)$ and $p(\theta|\mathcal{D}_i)$ belong to the same exponential family.

**Generalized Linear Model (GLM).** App. A.1 introduces GLMs and states that the polynomial approximation to the GLM mapping function is an exponential family model. Tempering the GLM likelihood function by $\kappa_i$ is equivalent to scaling the GLM mapping function by $\kappa_i$ and can achieved by scaling the polynomial approximate SS by the same factor.

## E.2 Smaller Tempering Factor Decreases Reward Value Proof

The KL divergence between two members of the same exponential family with natural parameters $\eta$ and $\eta'$, and log partition function $B(\cdot)$ is given by $(\eta - \eta')^\top \nabla B(\eta) - B(\eta) + B(\eta')$ [41]. To ease notational overload, we abuse some existing ones, which only apply in this subsection, by letting $\boldsymbol{s}_N \triangleq \sum_{k \in N} \boldsymbol{s}_k$ and $c_N \triangleq \sum_{k \in N} c_k$. Let $\eta'$ and $\eta$ be the natural parameters of the prior and the normalized tempered posterior distribution (used to generate a model reward with value $r_i$), respectively. Then, $\eta = \eta' + \kappa_i \left[ \boldsymbol{s}_N^\top, \ c_N \right]^\top$. For $\kappa_i \in [0,1]$, the derivative of $r_i$ w.r.t. $\kappa_i$ is non-negative:

$$
\begin{aligned}
\frac{\mathrm{d} r_i}{\mathrm{d} \kappa_i} &= \frac{\partial r_i}{\partial \eta} \frac{\partial \eta}{\partial \kappa_i} \\
&= \left( (\eta - \eta')^\top \nabla^2 B(\eta) + \nabla B(\eta) - \nabla B(\eta) \right) \left[ \boldsymbol{s}_N^\top, \ c_N \right]^\top \\
&= \left[ \kappa_i \boldsymbol{s}_N^\top, \ \kappa_i c_N \right] \nabla^2 B(\eta) \left[ \boldsymbol{s}_N^\top, \ c_N \right]^\top \\
&= \kappa_i \left[ \boldsymbol{s}_N^\top, \ c_N \right] \nabla^2 B(\eta) \left[ \boldsymbol{s}_N^\top, \ c_N \right]^\top \geq 0 .
\end{aligned}
$$

As $B(\eta)$ is convex w.r.t. $\eta$, the second derivative[21] $\nabla^2 B(\eta)$ is positive semi-definite, so $\left[ \boldsymbol{s}_N^\top, \ c_N \right] \nabla^2 B(\eta) \left[ \boldsymbol{s}_N^\top, \ c_N \right]^\top \geq 0$.

Hence, for $\kappa_i \in [0,1]$, the KL divergence is non-decreasing as $\kappa_i$ increases to 1. In other words, as $\kappa_i$ shrinks towards 0, the KL divergence is decreasing; equality only holds when the variance of the SS is 0.

## E.3 Implementation of Reward Control Mechanisms

This subsection introduces how to obtain the model reward $q_i(\theta)$ for each party $i$ in Sec. 5.

**Update for noise addition (varying $\tau_i$).** We update the inputs to Algorithm 1 for BLR or the No-U-Turn sampler for GLM. To generate party $i$'s posterior samples, for every party $k \in N$, the algorithm use the further perturbed SS $\boldsymbol{t}_k^i$ instead of the perturbed SS $\boldsymbol{o}_k$. Moreover, the algorithm consider the total DP noise $Z_k + \mathcal{N}(\boldsymbol{0}, 0.5\,\lambda\,\Delta_2^2(g_k)\,\tau_i\,\boldsymbol{I})$ instead of only the noise $Z_k$ added by party $k$.

**Update for likelihood tempering (varying $\kappa_i$).** To generate party $i$'s posterior samples, for every party $k \in N$, Use $\kappa_i c_k$, $\kappa_i \boldsymbol{o}_k$, and $p(\kappa_i Z_k)$ as the inputs to Algorithm 1 for BLR or the No-U-Turn [19] sampler for GLM instead. Scaling the perturbed SS would affect the sensitivity of party $k$'s submitted information and the DP noise needed.

# F   Time Complexity

---

**Algorithm 2** An overview of our collaborative ML problem setup.
The computational complexity is given in App. F.

---

**Require:** Rényi DP $\lambda$ parameter, Noise-aware inference algorithm, Shared prior $p(\theta)$ of model parameters and prior $p(\omega)$ of data parameters, $\rho$-Shapley fairness scheme parameter.

    *// Party's actions* *(ensure DP)*
1: **for** each party $i \in N$ **do**
2:     Compute exact SS $\boldsymbol{s}_i$ from dataset $\mathcal{D}_i$.
3:     Choose DP guarantee $(\lambda, \epsilon_i)$-Rényi DP.

---

[21]This second derivative is the variance of the sufficient statistic of $\theta$. It is non-negative and often positive.

4:     Sample $\boldsymbol{z}_i$ from the Gaussian distribution $p(Z_i) = \mathcal{N}(\boldsymbol{0},\ 0.5\ (\lambda/\epsilon_i)\ \Delta_2^2(g)\ \boldsymbol{I})$.
5:     Compute perturbed SS $\boldsymbol{o}_i \triangleq \boldsymbol{s}_i + \boldsymbol{z}_i$.
6:     Submit (i) number $c_i \triangleq |\mathcal{D}_i|$ of data points in its dataset $\mathcal{D}_i$, (ii) perturbed SS $\boldsymbol{o}_i$ and (iii) Gaussian distribution $p(Z_i)$ to the mediator.
7: **end for**

*// Mediator's actions*
*// 1. Compute valuation of perturbed SS needed for Shapley value. The choice of $v$ ensures a privacy-valuation trade-off.*
8: Draw $m$ samples from $p(\theta)$.
9: **for** each coalition $C \subseteq N$ **do**
10:     Draw $m$ samples from the posterior $p(\theta|\boldsymbol{o}_C)$ by applying the noise-aware inference algorithm. The algorithm requires the perturbed SS $\boldsymbol{o}_C \triangleq \{\boldsymbol{o}_i\}_{i \in C}$, data quantities $\{c_i\}_{i \in C}$ and noise distributions $\{Z_i\}_{i \in C}$.
11:     Compute $v_C$ by using the nearest-neighbors method [45] to estimate the KL divergence $D_{\text{KL}}(p(\theta|\boldsymbol{o}_C); p(\theta))$ from the samples.
12: **end for**

*// 2. Decide the target reward values using $\rho$-Shapley value [47] which ensure efficiency (P2), fairness (P3), rationality (P4) and control group welfare (P6).*
13: **for** each party $i \in N$ **do**
14:     Compute Shapley value $\phi_i = (1/n) \sum_{C \subseteq N \setminus i} \left[ \binom{n-1}{|C|}^{-1} \left( v_{C \cup \{i\}} - v_C \right) \right].$
15: **end for**
16: Identify the maximum Shapley value $\phi_* = \max_{k \in N} \phi_k$.
17: **for** each party $i \in N$ **do**
18:     Compute $\rho$-Shapley fair target reward $r_i^*$ for party $i$ using the formula $r_i^* = v_N \times (\phi_i/\phi_*)^\rho$
19: **end for**

*// 3. Generate model reward $q_i(\theta)$ with value $r_i = r_i^*$ that preserves similarity (P5) with the grand coalition's model and privacy for others (P1).*
20: **for** each party $i \in N$ **do**
21:     Initialize $\texttt{Kr} = ()$.
22:     **while** $\texttt{True}$ **do**
23:         Select $\kappa_i \in [0, 1]$ using a root finding algorithm and $\texttt{Kr}$.
24:         Draw $m$ samples from the normalized posterior $q_i(\theta)$ (Eq. 1 by applying the noise-aware inference algorithm. Use the scaled perturbed SS $\{\kappa_i \boldsymbol{o}_i\}_{i \in N}$, data quantities $\{\kappa_i c_i\}_{i \in N}$ and noise distributions $\{\kappa_i Z_i\}_{i \in N}$.
25:         Compute the reward value $r_i$ by using the nearest-neighbors method [45] to estimate the KL divergence $D_{\text{KL}}(q_i(\theta); p(\theta))$ from the samples.
26:         **if** attained reward value $r_i = r_i^*$ **then**
27:             Reward party $i$ with the $m$ posterior samples from $q_i(\theta)$.
28:             **break**
29:         **end if**
30:         Update $\texttt{Kr} \leftarrow \texttt{Kr} + ((\kappa_i, r_i),)$
31:     **end while**
32: **end for**

The main steps of our scheme are detailed in Algo. 2 and the time complexity of the steps are as follows:

1. **Local SS $\boldsymbol{s}_i$ computation (Line 2 in Algo 2).** Party $i$ will need to sum the SS for its $c_i$ data points. Subsequent steps will not depend on the number $c_i$ of data points. The (approximate) SS is usually an $\mathcal{O}(d^2)$ vector where $d$ is the number of regression model features. **Therefore, this step incurs $\mathcal{O}(c_i d^2)$ time.**

2. **Perturbed SS $\boldsymbol{o}_i$ computation (Lines 4-5 in Algo 2).** Each party will need to use the Gaussian mechanism to perturb $\boldsymbol{s}_i$. **Therefore, this step incurs $\mathcal{O}(d^2)$ time.**



A — **Valuation of Perturbed SS (Sec. 3).** The valuation of $\boldsymbol{o}_N$ requires us to draw $m$ posterior samples of model parameters using DP noise-aware inference (refer to App. A.3 and the cited references for the exact steps). As the methods of [27] and [4] incur $\mathcal{O}(md^4)$ time for a single party, inference based on Fig. 4 will take $n$ times longer to consider $n$ parties. KL estimation using $k$-nearest neighbor will incur $\mathcal{O}(m\log(m)\dim(\theta))$ time to value multiple (scaled) perturbed SS. **Therefore, valuation incurs $\mathcal{O}(nmd^4 + m\log(m)\dim(\theta))$ time**.



3. **Deciding target reward value $r_i^*$ for every $i \in N$ (Sec. 4, Lines 9-19 in Algo 2)).** Computing the Shapley values exactly involves valuing $\boldsymbol{o}_C$ for each subset $C \subseteq N$, hence, repeating Step A $\mathcal{O}(2^n)$ time. When the number of parties is small (e.g., $< 6$), we can compute the Shapley values exactly. For larger $n$, we can approximate the Shapley values $(\phi_i)_{i \in N}$ with bounded $\ell_2$-norm error using $\mathcal{O}(n(\log n)^2)$ samples [25, 53]. Moreover, the value of different coalitions can be computed in *parallel*. **Therefore, this step incurs $\mathcal{O}(2^n)$ or $\mathcal{O}(n(\log n)^2)$ times the time in Step A**.

4. **Solving for $\kappa_i$ to generate model reward (Sec. 5.2, Lines 21-31 in Algo 2).** During root-finding, the mediator values different model rewards $q_i(\theta)$ generated by scaling the perturbed SS $\boldsymbol{o}_k$, data quantity $c_k$ and DP noise distribution $Z_k$ of each party $k \in N$ by different $\kappa_i$, hence, repeats Step A. As we are searching for the root in a fixed interval $[0, 1]$ and to a fixed precision, Step A is repeated a *constant* (usually $< 10$) number of times. **Therefore, this step incurs $\mathcal{O}(nmd^4 + m\log(m)\dim(\theta))$ time per party**.

   The mediator can further reduce the number of valuation of model rewards (repetitions of Step A) by using the tuples of $(\kappa_i, r_i)$ obtained when solving for $\kappa_i$ to narrow the root-finding range for other parties after $i$.

Therefore, the total incurred time depends on the number of valuations performed in Step A. The time complexity may vary for other inference and KL estimation methods.

## G Comparison of Reward Control Mechanisms via Noise Addition (Sec. 5.1) vs. Likelihood Tempering (Sec. 5.2)

See Fig. 7.

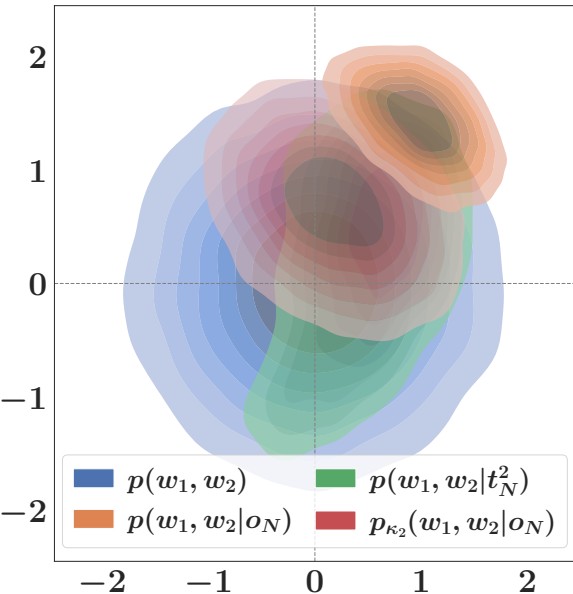

Figure 7: We contour plot the distribution of the regression model weights $w_1$ and $w_2$ for the prior, the grand coalition $N$'s posterior, and the model reward's posterior to attain the target reward value $r_2^*$ utilizing noise addition (Sec. 5.1) vs. likelihood tempering (Sec. 5.2) as the reward control mechanism for the Syn dataset where $\rho = .5$. The tempered posterior interpolates the prior and grand coalition $N$'s posterior better as its mean/mode lies along the line connecting the prior's and grand coalition $N$'s posterior mean and the variance is scaled by the same extent for both weights.

## H Experiments

The experiments are performed on a machine with Ubuntu 20.04 LTS, $2\times$ Intel Xeon Gold 6230 (2.1GHz) without GPU. The software environments used are Miniconda and Python. A full list of packages used is given in the file environment.yml attached.

### H.1 Experimental Details

**Synthetic BLR (Syn).** The BLR parameters $\theta$ consist of the weights for each dimension of the 2D dataset, the bias, and the variance $\sigma^2$. The *normal inverse-gamma* distribution used (i) to generate the true regression model weights, variance, and a 2D dataset and (ii) as our model prior is as follows: $\sigma^2 \sim \texttt{InvGamma}(\alpha_0 = 5, \beta_0 = 0.1)$ where $\alpha_0$ and $\beta_0$ are, respectively, the inverse-gamma shape and scale parameters, and $\boldsymbol{w}|\sigma^2 \sim \mathcal{N}(\boldsymbol{0}, \sigma^2\boldsymbol{\Lambda}_0^{-1})$ where $\boldsymbol{\Lambda}_0 = 0.025\,\boldsymbol{I}$.

We consider three parties 1, 2, and 3 with $c_0 = 100$, $c_1 = 200$, and $c_2 = 400$ data points, respectively. We fix $\epsilon_1 = \epsilon_3 = 0.2$ and vary $\epsilon_2$ from the default 0.1. As $\epsilon_2$ increases (decreases), party 2 may become the most (least) valuable. We allow each party to have a different Gaussian distribution $p(\boldsymbol{x}_i)$ by maintaining a separate conjugate *normal inverse-Wishart* distribution $p(\omega_i = (\mu_{\omega,i}, \Sigma_{\omega,i}))$ for each party. We set the prior $\Sigma_{\omega,i} \sim \mathcal{W}^{-1}(\psi_0 = \boldsymbol{I}, \nu_0 = 50)$ where $\psi_0$ and $\nu_0$ are the scale matrix and degrees of freedom (i.e., how strongly we believe the prior), respectively. Then, $\mu_{\omega,i} \sim \mathcal{N}(\boldsymbol{0}, \Sigma_{\omega,i})$. The $\ell_2$-sensitivity is estimated using [26]'s analysis based on the norms/bounds of the dataset.

One posterior sampling run generates 16 Gibbs sampling chains in parallel. For each chain, we discard the first 10000 burn-in samples and draw $m = 30000$ samples. To reduce the closeness/correlation between samples which will affect the nearest-neighbor-based KL estimation, we thin and only keep

every 16-th sample and concatenate the thinned samples across all 16 chains. For the experiment on reward control mechanisms, we use 5 independent runs of posterior sampling and KL estimation.

**Californian Housing dataset (CaliH).**   As the CaliH dataset may contain outliers, we preprocess the "public" dataset (60% of the CaliH data) by subtracting the median and scaling by the interquartile range for each feature. We train a *neural network* (NN) of 3 layers with $[48, 24, 6]$ hidden units and the *rectified linear unit* (ReLU) as the activation function to minimize the mean squared error, which we will then use as a "pre-trained NN". The outputs of the last hidden layer have 6 features used as the inputs for BLR. We intentionally reduce the number of features in the BLR model by adding more layers to the pre-trained NN and reduce the magnitude of the BLR inputs by adding an activation regularizer on the pre-trained NN hidden layers (i.e., $\ell_2$ penalty weight of 0.005). These reduce the computational cost of Gibbs sampling/KL estimation and the $\ell_2$-sensitivity of the inputs to BLR (hence the added DP noise), respectively. We also add a weights/bias regularizer with an $\ell_2$ penalty weight of 0.005 for the last layer connected to the outputs. Lastly, we standardize the outputs of the last hidden layer to have zero mean and unit variance.

We preprocess the private dataset for valuation and the held-out validation set (an 80-20 split) using the same pre-trained NN/transformation process. To reduce the sensitivity and added DP noise, we filter and exclude any data point with a $z$-score $> 4$ for any feature. There are 6581 training data points left. We divide the dataset randomly among 3 parties such that parties 1, 2, and 3 have, respectively, $20\%, 30\%$ and $50\%$ of the dataset and $\epsilon_1 = \epsilon_3 = 0.2$ while $\epsilon_2$ is varied from the default 0.1.

The BLR parameters $\theta$ consist of the weights for each of the 6 features, the bias, and the variance $\sigma^2$. We assume $\theta$ has a *normal inverse-gamma* distribution and set the prior as follows. The prior depends on the MLE estimate based on the public dataset, and we assume it has the same significance as $n_0 = 10$ data points. Hence, we set $\sigma^2 \sim \texttt{InvGamma}(\alpha_0 = n_0/2, \beta_0 = n_0/2 \times \text{MLE estimate of } \sigma^2)$ and $\boldsymbol{w}|\sigma^2 \sim \mathcal{N}(\boldsymbol{0}, \sigma^2(n_0\,\boldsymbol{x}^\top\boldsymbol{x})^{-1})$.

We assume that each party has the same underlying Gaussian distribution for $p(\boldsymbol{x})$ and maintain only one conjugate *normal inverse-Wishart* distribution $p(\omega = (\mu_\omega, \Sigma_\omega))$ shared across parties. We initialize the prior $p(\omega)$ to be weakly dependent on the prior dataset [39]. The $\ell_2$-sensitivity is estimated using [26]'s analysis based on the norms/bounds of the private transformed dataset.

One posterior sampling run generates 16 Gibbs sampling chains in parallel. For each chain, we discard the first 10000 burn-in samples and draw $m = 30000$ samples. To reduce the closeness/correlation between samples which will affect the nearest-neighbor-based KL estimation, we thin and only keep every 16-th sample and concatenate the thinned samples across all 16 chains. For the experiment on reward control mechanisms, we use 5 independent runs of posterior sampling and KL estimation.

**PIMA Indian Diabetes classification dataset (Diab).**   This dataset has 8 raw features such as age, BMI, number of pregnancies, and a binary output variable. Patients with and without diabetes are labeled $y = 1$ and $y = -1$, respectively. We split the training and the validation set using an 80-20 split. There are 614 training data points. There are $35.6\%$ and $31.8\%$ of patients with diabetes in the training and validation sets, respectively. Hence, random guessing would lead to a cross-entropy loss of 0.629.

We preprocess both sets by (i) subtracting the training set's median and scaling by the interquartile range for each feature, (ii) using *principal component analysis* (PCA) to select the 4 most important components of the feature space to be used as new features, and lastly, (iii) centering and scaling the new features to zero mean and unit variance. To reduce the effect of outliers and the $\ell_2$-sensitivity, we clip each training data point's feature values at $\pm 2.2$.

We divide the 614 training data points such that parties 1, 2, and 3 have, respectively, $20\%, 30\%$, and $50\%$ of the dataset and $\epsilon_1 = \epsilon_3 = 0.2$ while $\epsilon_2$ is varied from the default 0.1. We compute the approximate SS [21] and perturb them for each party to achieve the selected $\epsilon_i$ [27]. The $\ell_2$-sensitivity is also estimated based on the dataset.

We consider a Bayesian logistic regression model, and its parameters $\theta$ consist of the bias and the weights for each of the 4 features. Like that of [27], we set an independent standard Gaussian prior for $\theta$ but rescale it such that the squared norm $\|\theta\|_2^2$ has a truncated Chi-square prior with $d = 4$ degrees

of freedom. Truncation prevents sampling $\theta$ with a norm larger than 2.5 times the non-private/true setting's $\theta^*$ squared norm during inference.

We assume each party has the same distribution for $p(\boldsymbol{x})$. Our data prior $p(\boldsymbol{x})$ has mean $\mathbf{0}$ and covariance $\Sigma = \text{diag}(\iota) \, \Omega \, \text{diag}(\iota)$ where $\iota \sim \mathcal{N}(\mathbf{1}, \boldsymbol{I}), \iota \in [0, 2]$, and $\Omega$ follows a Lewandowski-Kurowicka-Joe prior LKJ(2).

We use the *No-U-Turn* [19] sampler. We run 25 Markov chains with 400 burn-in samples and draw $m = 2000$ samples with a target Metropolis acceptance rate of 0.86. We discard chains with a low Bayesian fraction of missing information (i.e., $< .3$) and split the concatenated samples across chains into 5 groups to estimate KL divergence. As sampling is slower and the generated samples tend to be less correlated, we can use fewer samples.

*Remark.* For the CaliH dataset, the preprocessing is based on the "public" dataset, but for the Diab dataset, the preprocessing (i.e., standardization, PCA) is based on the private, valued dataset. We have assumed that the data is preprocessed. However, in practice, before using our mechanism, the parties may have to reserve/separately expend some privacy budget for these processing steps. The privacy cost is ignored in our analysis of the privacy-valuation trade-off.

**KL estimation.**  We estimate KL divergence using the $k$-nearest-neighbor-based KL estimator [45]. To reduce the bias due to the skew of the distribution, we apply a whitening transformation [54] where each parameter sample is centered and multiplied by the inverse of the sample covariance matrix based on all samples from $\theta$ and $\theta \mid \boldsymbol{o}$. As a default, we set $k = 4$ and increase $k$ until the distance to the $k$-th neighbor is non-zero.

## H.2   Utility of Model Reward

The *mean negative log probability* (MNLP) on a test dataset $\mathcal{D}_*$ given the perturbed SS $\boldsymbol{o}_i$ is defined as follows:
$$\text{MNLP} \triangleq \frac{1}{|\mathcal{D}_*|} \sum_{(\boldsymbol{x}_*, y_*) \in \mathcal{D}_*} -\log p(y_*|\boldsymbol{x}_*, \boldsymbol{o}_i) \ .$$

We prefer MNLP over the *model accuracy* or *mean squared error* metric. MNLP additionally measures if a model is uncertain of its accurate predictions or overconfident in inaccurate predictions. In contrast, the latter metrics penalize inaccurate predictions equally and ignore the model's confidence (which is affected by the DP noise).

**Regression.**  Approximating the predictive distribution, $p(y_*|\boldsymbol{x}_*, \boldsymbol{o}_i)$, for test input $\boldsymbol{x}_*$ as Gaussian, the MNLP formula becomes

$$\text{MNLP} \triangleq \frac{1}{|\mathcal{D}_*|} \sum_{(\boldsymbol{x}_*, y_*) \in \mathcal{D}_*} \frac{1}{2} \left( \log(2\pi\widehat{\sigma^2}(\boldsymbol{x}_*)) + \frac{(\widehat{\mu}(\boldsymbol{x}_*) - y_*)^2}{\widehat{\sigma^2}(\boldsymbol{x}_*)} \right)$$

where $\mu_*$ and $\widehat{\sigma^2}(\boldsymbol{x}_*)$ denote the predictive mean and variance, respectively. The first term penalizes large predictive variance while the second term penalizes inaccurate predictions more strongly when the predictive variance is small (i.e., overconfidence).

- The predictive mean $\widehat{\mu}(\boldsymbol{x}_*)$ is the averaged prediction of $\boldsymbol{y}_*$ (i.e., $\boldsymbol{w}^\top \boldsymbol{x}_*$, where $\boldsymbol{w}$ is part of the model parameters $\theta$) over all samples of the model parameters $\theta$.
- The predictive variance $\widehat{\sigma^2}(\boldsymbol{x}_*)$ is computed using the variance decomposition formula, i.e., the sum of the averaged $\sigma^2$ (the unknown variance parameter within $\theta$) and the computed variance in predictions over samples, i.e., $= m^{-1} \sum_{j=1}^{m} \sigma_j^2 + \widehat{\mu^2}(\boldsymbol{x}_*) - \widehat{\mu}(\boldsymbol{x}_*)^2$.

**Classification.**  We can estimate $p(y_*|\boldsymbol{x}_*, \boldsymbol{o}_i)$, for test input $\boldsymbol{x}_*$ using the Monte Carlo approximation [39] and reusing the samples $\theta$ from $p(\theta|\boldsymbol{o}_i)$. Concretely, $p(y = 1|\boldsymbol{x}_*, \boldsymbol{o}_i) \approx m^{-1} \sum_{j=1}^{m} \sigma(\theta^\top \boldsymbol{x}_*)$. The MNLP is equivalent to the cross-entropy loss.

## H.3   Baselines

To plot the figures in Sec. 6, the baseline DP and collaborative ML works must

1. work for similar models, i.e., Bayesian linear and logistic regression;
2. not use additional information to value coalitions and generate model rewards (to preserve the DP post-processing property); and
3. decide feasible model reward values and suggest how model rewards can be generated.

**Work of [59].** Valuation by volume is model-agnostic (satisfying criteria 1). Each party $i \in N$ can submit the noisy version of $\boldsymbol{X}_i^\top \boldsymbol{X}_i$ with DP guarantees to the mediator who can sum them to value coalitions (satisfying criteria 2). The work does not propose a model reward scheme to satisfy criteria 3.

**Work of [47].** [47] only considered Bayesian linear regression (with known variance) and it is not straightforward to compute information gain on model parameters for Bayesian linear regression (with unknown variance) and Bayesian logistic regression. Thus, the work does not satisfy criteria 1. For Bayesian linear regression (with known variance), each party $i \in N$ can submit the noisy version of $\boldsymbol{X}_i^\top \boldsymbol{X}_i$ with DP guarantees to the mediator who can sum them to value coalitions (satisfying criteria 2). The work proposed a model reward scheme which involves adding noise to the outputs $\boldsymbol{y}$ ( satisfying criteria 3 but has to be adapted to ensure DP).

**DP-FL works.** A promising approach is to use `DP-FedAvg/DP-FedSGD` [36] to learn any model parameters (satisfying criteria 1) in conjunction with `FedSV` [55] to value coalitions without additional information (satisfying criteria 2). However, to our knowledge, these works will not satisfy criteria 3 as they do not suggest how to generate model rewards of a target reward value.

As no existing work satisfies all criteria, we compare against (1) using non-noise-aware inference instead of noise-aware inference, all else equal (in App. H.5); and (2) an adapted variant of the reward control via noise addition (in Sec. 5.1, Sec. 6, and App. G).

## H.4 Valuation Function

In Sec. 6, we only vary the privacy guarantee $\epsilon_i$ of one party $i$. In this subsection, we will analyze how other factors such as the coverage of the input space and the number of posterior samples on the valuation $v_i$.

**Coverage of input space.** We vary the coverage of the input space by only keeping those data points whose first feature value is not greater than the $25, 50, 75, 100$-percentile. Across all experiments in Table 1, it can be observed that as the percentile increases (hence, data quantity and coverage improve), the surprise elicited by the perturbed SS $\boldsymbol{o}_N$ increases in tandem with the surprise elicited by the exact SS $\boldsymbol{s}_N$.

| Feature 0's Percentile Range | $[0, 25]$ | $[0, 50]$ | $[0, 75]$ | $[0, 100]$ |
|---|---|---|---|---|
| **Syn** | | | | |
| Surprise of $\boldsymbol{s}_N$ | 12.030 | 12.698 | 13.322 | 14.183 |
| Surprise of $\boldsymbol{o}_N$ | 6.007 | 6.775 | 7.410 | 8.438 |
| **CaliH** | | | | |
| Surprise of $\boldsymbol{s}_N$ | 21.261 | 22.401 | 26.578 | 28.422 |
| Surprise of $\boldsymbol{o}_N$ | 9.282 | 10.212 | 12.121 | 17.959 |
| **Diab** | | | | |
| Surprise of $\boldsymbol{s}_N$ | 5.450 | 6.279 | 7.019 | 7.258 |
| Surprise of $\boldsymbol{o}_N$ | 1.854 | 2.712 | 3.909 | 5.394 |

Table 1: We report the surprise elicited by $\boldsymbol{s}_N$ and $\boldsymbol{o}_N$ (with $\epsilon = 1$) when using the subset of data with first feature value not exceeding the $25, 50, 75, 100$-percentile for all datasets.

**Number of posterior samples.** For a consistent KL estimator, the bias/variance of the KL estimator should decrease with a larger number of posterior samples.

**Gibbs sampling.** We compare the estimated surprise using various degrees of thinning (i.e., keeping only every $t$-th sample) to generate 30000 samples for the CaliH dataset. In Table 2, it can be observed that although the total number of samples is constant, the surprise differs significantly. Moreover,

as $t$ increases, the surprise decreases at a decreasing rate and eventually converges. This may be because consecutive Gibbs samples are highly correlated and close, thus causing us to underestimate the distance to the $k$-th nearest-neighbors within $\theta \mid o_N$ (see discussion in App. C.3). Increasing $t$ reduces the correlation and closeness and better meets the i.i.d. samples assumption of the nearest-neighbor-based KL estimation method [45].

| Thin every $t$-th sample | Surprise $v_N$ |
|---:|---|
| 1 | $14.849 \pm 0.036$ |
| 2 | $12.839 \pm 0.033$ |
| 4 | $11.626 \pm 0.018$ |
| 8 | $11.038 \pm 0.022$ |
| 16 | $10.834 \pm 0.033$ |
| 20 | $10.790 \pm 0.032$ |
| 30 | $10.793 \pm 0.011$ |

Table 2: Thinning factor $t$ vs. surprise $v_N$ for CaliH dataset.

**NUTS logistic regression.** After drawing 10000 samples for the Diab dataset using the default setting, we analyze how using a subset of the samples will affect the estimated surprise. In particular, we consider using (i) the *first* $m$ samples or (ii) *thinned* $m$ samples where we only keep every $10000/m$-th sample.

In Table 3, it can be observed that as the number $m$ of samples increases, the standard deviation of the estimated surprise decreases. Moreover, there is no significant difference between using the first $m$ samples or the thinned $m$ samples. This suggests that the samples are sufficiently independent and thinning is not needed.

| No. $m$ of Samples | Surprise $v_N$ |
|---:|---|
| First 1000 | $2.227 \pm 0.051$ |
| Thinned 1000 | $2.211 \pm 0.034$ |
| First 2000 | $2.117 \pm 0.049$ |
| Thinned 2000 | $2.117 \pm 0.045$ |
| First 5000 | $2.145 \pm 0.037$ |
| Thinned 5000 | $2.119 \pm 0.038$ |
| All 10000 | $2.128 \pm 0.030$ |

Table 3: Number $m$ of samples vs. surprise $v_N$ for Diab dataset.

### H.5 Additional Experiments on Valuation, Privacy-valuation Trade-off, and Privacy-reward Trade-off

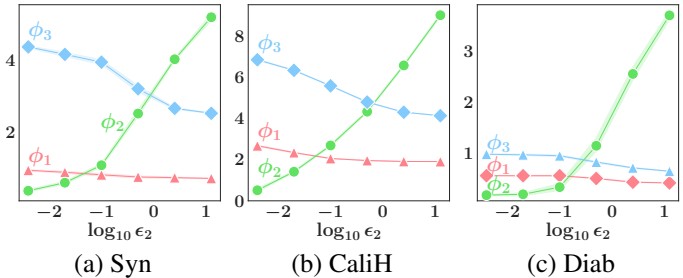

(a) Syn      (b) CaliH      (c) Diab

Figure 8: Graphs of Shapley value $\phi_i$ of parties $i = 1, 2, 3$ vs. party 2's privacy guarantee $\epsilon_2$ for various datasets.

**Shapley value.** In Fig. 8, it can be observed that as party 2 weakens its privacy guarantee (i.e., $\epsilon_2$ increases), its Shapley value $\phi_2$ increases while other parties' Shapley values (e.g., $\phi_3$) decrease.

When party 2 adds less noise to generate its perturbed SS $\boldsymbol{o}_2$, others add less value (i.e., make lower *marginal contributions* (MC)) to coalitions containing party 2. Party 2 changes from being least valuable to being most valuable, even though it has more data than party 1 and less data than party 3.

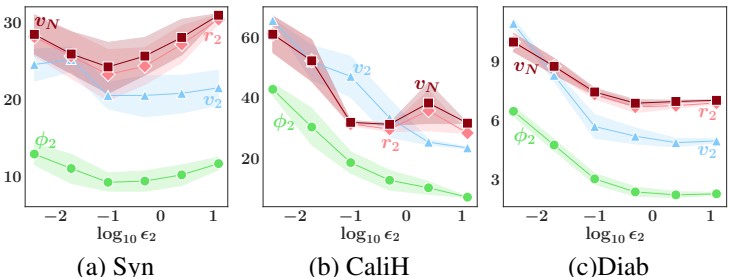

(a) Syn            (b) CaliH            (c)Diab

Figure 9: Graphs of party 2's valuation $v_2$, Shapley value $\phi_2$, and attained reward value $r_2$ vs. privacy guarantee $\epsilon_2$ for various datasets when performing non-noise-aware (i.e., noise-naive) inference, i.e., $p(\theta|S_N = \boldsymbol{o}_N)$ and treating $\boldsymbol{o}_N$ as though it is $\boldsymbol{s}_N$.

**Without DP noise-aware inference.** In Fig. 9a, it can be observed that as $\epsilon_2$ increases, $v_i$ and $\phi_i$ for party $i = 2$ do not strictly increase. In Figs. 9b-c, it can be observed that as $\epsilon_2$ increases, $v_i$ and $\phi_i$ for party $i = 2$ decrease instead. The consequence of non-noise-aware inference is undesirable for incentivization — party 2 unfairly gets a lower valuation and reward for using a weaker privacy guarantee, i.e., a greater privacy sacrifice. Moreover, when $\epsilon_2$ is small (i.e., under a strong privacy guarantee), party 2 is supposed to be least valuable. However, the significantly different $\boldsymbol{o}_2$ causes party 2 to have the highest valuation and be rewarded with the grand coalition $N$'s model (i.e., $r_i$ close to $v_N$) instead.

Lastly, we also observe that without DP noise-aware inference, the utility of the model reward is small. For example, the naive posterior for the Syn dataset results in an MNLP larger than 100.

**Conditions for larger improvement in MNLP.** In Fig. 3, it seems that the utility of party $i = 2$'s model reward measured by $\text{MNLP}_r$ cannot improve significantly over over that of its individually trained model when $\epsilon_2$ is large. However, party $i$'s $\text{MNLP}_r$ can be improved by a larger extent when (i) any other party $j \neq i$ selects a weaker privacy guarantee (i.e., a larger $\epsilon_j$), thus improving the utility of the collaboratively trained model or (ii) party $i$ and others have lower data quantity (i.e., smaller $c_k$ for all $k \in N$) and are unable to individually train a model of high utility. Figs. 10a, 10b, and 10c are examples of (i), (ii), and (i+ii), respectively. In Fig. 10a, the $\text{MNLP}_N$ of grand coalition $N$'s collaboratively trained model is lower than that in Fig. 3a. In Fig. 10b, the $\text{MNLP}_i$ of party $i$'s model is higher due to less data. In these examples, we observe that a party can still gain a significant improvement $\text{MNLP}_i - \text{MNLP}_r$ when $\epsilon_i > 1$.

Condition (i) for a larger improvement in $\text{MNLP}_r$ is satisfied when the trade-off deters parties from selecting excessive DP guarantees, i.e., it incentivizes parties to select weaker DP guarantees that still meet their legal and customers' requirements. Condition (ii) should be satisfied in most real-life scenarios where a party wants to participate in collaborative ML and federated learning. The party (e.g., bank) is unable to achieve its desired utility with its individually trained model due to limited data and collaborates with others to unlock any improvement in the utility of a collaboratively trained model.

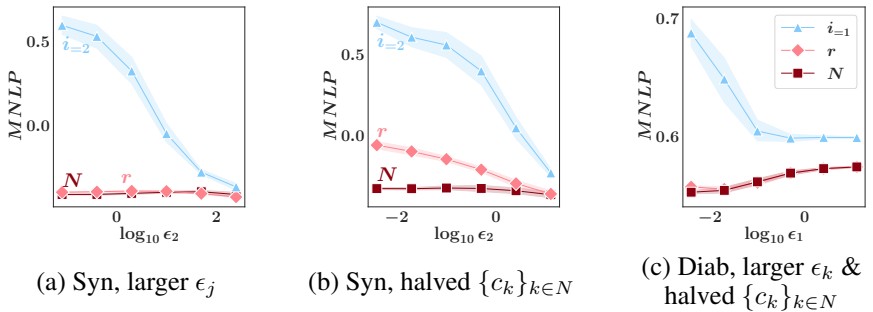

(a) Syn, larger $\epsilon_j$     (b) Syn, halved $\{c_k\}_{k \in N}$     (c) Diab, larger $\epsilon_k$ & halved $\{c_k\}_{k \in N}$

Figure 10: Graphs of utility of party $i = 2$'s model reward $q_i(\theta)$ measured by MNLP$_r$ vs. privacy guarantee $\epsilon_2$ for Syn dataset (a) when $\epsilon_1 = \epsilon_3 = 2$ instead of $0.2$, and (b) when only a subset of $c_k/2$ data points is available for every party $k = 1, 2, 3$. (c) Graph of utility of party $i = 1$'s model reward $q_i(\theta)$ measured by MNLP$_r$ vs. privacy guarantee $\epsilon_1$ for Diab dataset when $\epsilon_2 = \epsilon_3 = 2$ instead of $0.2$ and only a subset of $c_k/2$ data points is available for every party $k = 1, 2, 3$.

**Higher $\lambda = 10$.** In Fig. 11, the privacy-valuation, privacy-reward, and privacy-utility trade-offs are still observed when parties select a higher $\lambda = 10$ when enforcing the Rényi DP guarantee. Moreover, the utility of party 2's model reward is higher (i.e., lower MNLP) than that of its individually trained model.

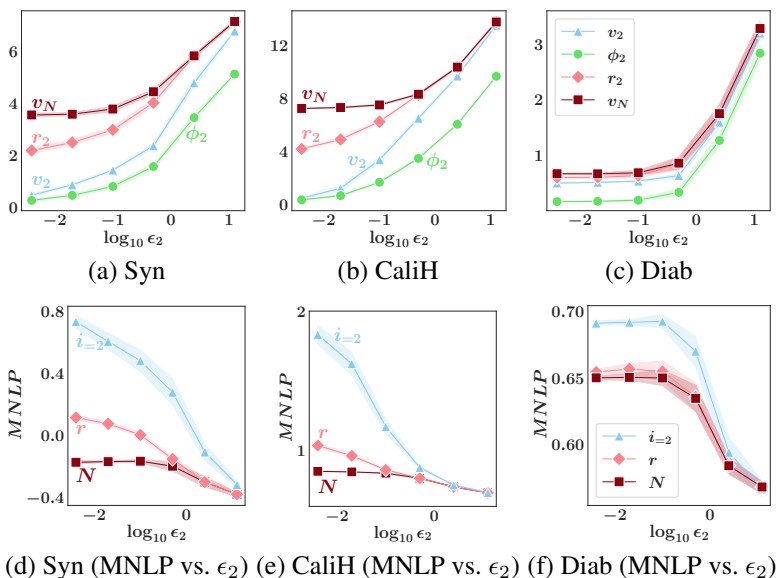

(a) Syn     (b) CaliH     (c) Diab

(d) Syn (MNLP vs. $\epsilon_2$) (e) CaliH (MNLP vs. $\epsilon_2$) (f) Diab (MNLP vs. $\epsilon_2$)

Figure 11: Graphs of party 2's (a-c) valuation $v_2$, Shapley value $\phi_2$, and attained reward value $r_2$, and (d-f) utility of its model reward $q_i(\theta)$ measured by MNLP$_r$ vs. privacy guarantee $\epsilon_2$ for various datasets when enforcing $(\lambda = 10, \epsilon_i)$-Rényi DP guarantee.

## H.6 Additional Experiments on Reward Control Mechanisms

For the CaliH dataset, there is a monotonic relationship between $r_i$ vs. both $\kappa_i$ and $\tau_i$, as shown in Fig. 12a. However, it can be observed from Figs. 12b-c that for the same attained reward value $r_i$, adding scaled noise variance $\tau_i$ will lead to a lower similarity $r_i'$ to the grand coalition $N$'s posterior $p(\theta|\boldsymbol{o}_N)$ and utility of model reward (higher MNLP$_r$) than tempering the likelihood by $\kappa_i$.

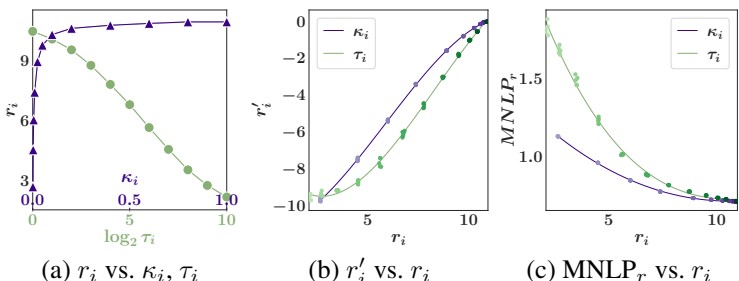

(a) $r_i$ vs. $\kappa_i, \tau_i$     (b) $r_i'$ vs. $r_i$     (c) MNLP$_r$ vs. $r_i$

Figure 12: (a) Graph of attained reward value $r_i$ vs. $\kappa_i$ (Sec. 5.2) and $\tau_i$ (Sec. 5.1), (b) graph of similarity $r_i'$ to the grand coalition $N$'s posterior $p(\theta|\boldsymbol{o}_N)$ vs. $r_i$, and (c) graph of utility of party $i = 2$'s model reward $q_i(\theta)$ measured by MNLP$_r$ vs. $r_i$ for CaliH dataset.

For Diab dataset, there is a monotonic relationship between $r_i$ vs. both $\kappa_i$ and $\tau_i$, as shown in Fig. 13a. However, it can be observed from Fig. 13b-c that for the same attained reward value $r_i$, tempering the likelihood by $\kappa_i$ leads to a higher similarity $r_i'$ to the grand coalition $N$'s posterior $p(\theta|\boldsymbol{o}_N)$ and utility of model reward (lower MNLP$_r$) than adding scaled noise variance $\tau_i$.

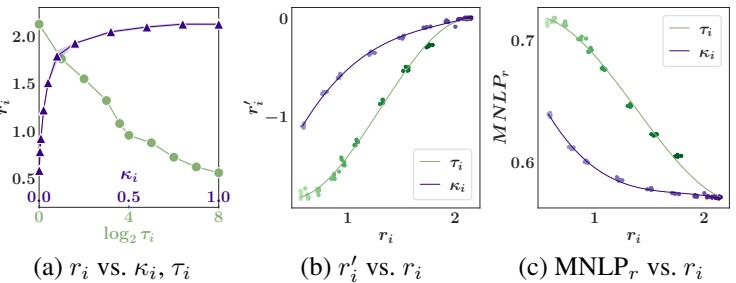

(a) $r_i$ vs. $\kappa_i, \tau_i$     (b) $r_i'$ vs. $r_i$     (c) MNLP$_r$ vs. $r_i$

Figure 13: (a) Graph of attained reward value $r_i$ vs. $\kappa_i$ (Sec. 5.2) and $\tau_i$ (Sec. 5.1), (b) graph of similarity $r_i'$ to the grand coalition $N$'s posterior $p(\theta|\boldsymbol{o}_N)$ vs. $r_i$, and (c) graph of utility of party $i = 2$'s model reward $q_i(\theta)$ measured by MNLP$_r$ vs. $r_i$ for Diab dataset.

**Problematic noise realization.** We will show here and in Fig. 14a that some (large) noise realization can result in a non-monotonic relationship between the attained reward value $r_i$ vs. the scaled additional noise variance $\tau_i$. As a result, it is hard to bracket the smallest root $\tau_i$ that solves for $r_i = r_i^*$ (e.g., $= 2$ or $= 3$). Moreover, it can be observed from Figs. 14b-c that the model reward's posterior $q_i(\theta)$ has a low similarity $r_i'$ to the grand coalition $N$'s posterior $p(\theta|\boldsymbol{o}_N)$ and a much higher MNLP$_r$ than the prior. This suggests that injecting noise does not interpolate well between the prior and the posterior. In these cases, it is not suitable to add scaled noise variance $\tau_i$ and our reward control mechanism via likelihood tempering with $\kappa_i$, is preferred instead.

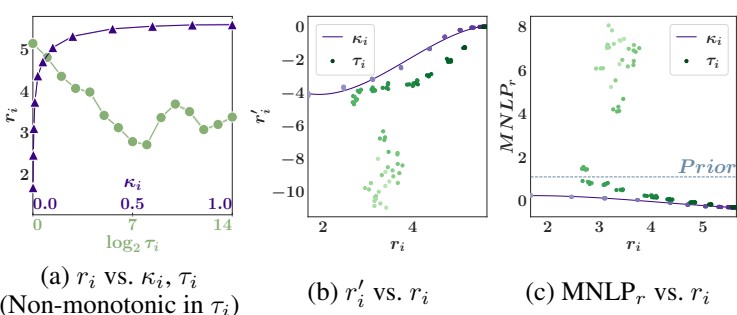

(a) $r_i$ vs. $\kappa_i, \tau_i$
(Non-monotonic in $\tau_i$)     (b) $r_i'$ vs. $r_i$     (c) MNLP$_r$ vs. $r_i$

Figure 14: (a) Graph of attained reward value $r_i$ vs. $\kappa_i$ (Sec. 5.2) and $\tau_i$ (Sec. 5.1), (b) graph of similarity $r_i'$ to the grand coalition $N$'s posterior $p(\theta|\boldsymbol{o}_N)$ vs. $r_i$, and (c) graph of utility of party $i = 2$'s model reward $q_i(\theta)$ measured by MNLP$_r$ vs. $r_i$ for Syn dataset corresponding to (a).

# I Other Questions

**Question 1: Are there any ethical concerns we foresee with our proposed scheme?**

**Answer:** Our privacy-valuation trade-off (V3) should deter parties from *unfetteredly* selecting excessively strong DP guarantees. Parties inherently recognize the benefits of stronger DP guarantees and may prefer such benefits in collaboration out of overcaution, mistrust of others, and convenience. The trade-off counteracts (see Fig. 1) the above perceived benefits by explicitly introducing costs (i.e., lower valuation and quality of model reward). Consequently, parties will carefully select a weaker yet satisfactory privacy guarantee they truly need.

However, a potential concern is that parties may opt to sacrifice their data's privacy to obtain a higher-quality model reward. The mediator can alleviate this concern by enforcing a minimum privacy guarantee (i.e., maximum $\epsilon$) each party must select. The model rewards will preserve this minimum privacy guarantee due to P1. The mediator can also decrease the incentive by modifying $v_C$.

Another potential concern is that if parties have data with significantly different quantity/quality/privacy guarantees, the weaker party $k$ with fewer data or requiring a stronger privacy guarantee will be denied the best model reward (i.e., trained on the grand coalition's SS) and instead rewarded with one that is of lower quality for fairness. The mediator can alleviate the concern and at least ensure individual rationality (P4) by using a smaller $\rho$ so that a weaker party $k$ can obtain a higher-quality model reward with a higher target reward value $r_k^*$.

**Question 2: Is it sufficient and reasonable to value parties based on the submitted information $\{c_i, o_i, p(Z_i)\}_{i \in N}$ instead of ensuring and incentivizing truthfulness? Would parties strategically declare other values to gain a higher valuation and reward?**

**Answer:** An ideal collaborative ML scheme should additionally incentivize parties to be truthful and verify the authenticity of the information provided. However, achieving the "truthfulness" incentive is hard and has only been tackled by existing works to a limited extent. Existing work cannot discern if the data and information declared are collected or artificially created (e.g., duplicated) and thus, this non-trivial challenge is left to future work. The work of [32] assigns and considers each client's reputation from earlier rounds, while the works of [29, 31] measure the correlation in parties' predictions and model updates. The work of [7] proposes a payment rule based on the log pointwise mutual information between a party's dataset and the pooled dataset of others. This payment rule guarantees that when all other parties are truthful (i.e., a strong assumption), misreporting a dataset with an inaccurate posterior is worse (in expectation) than reporting a dataset with accurate posterior.[22]

Thus, like the works of [18, 17, 25, 40, 47, 51] and others, we value data *as-is* and leave achieving the "truthfulness" incentive to future work. In practice, parties such as hospitals and firms will truthfully share information as they are primarily interested in building and receiving a model reward of high quality and may additionally be bound by the collaboration's legal contracts and trusted data-sharing platforms like Ocean Protocol [43]. For example, with the use of X-road ecosystem,[23] parties can maintain a private database which the mediator can query for the perturbed SS. This ensures the authenticity of the data (also used by the owner) and truthful computation given the uploaded private database.

Lastly, a party $k$ who submits fake SS will also reduce its utility from the collaboration. Party $k$'s fake SS will affect the grand coalition's posterior of the model parameters given all perturbed SS and is also used to generate $k$'s model reward. As party $k$ only receives posterior samples, $k$ cannot replace the fake SS with its exact SS locally. As party $k$ have to bear the consequences of the fake SS, it would be more likely to submit true information.

**Question 3: Why do we only consider Bayesian models with SS?**

---

[22]The payment rule may be unfair as when two parties are present, they will always be paid equally.

[23]https://joinup.ec.europa.eu/collection/ict-security/solution/
x-road-data-exchange-layer/about, https://x-road.global/

**Answer:** See App A.1 for a background on SS. Our approach would also work for Bayesian models with approximate SS, such as Bayesian logistic regression, and latent features extracted by a neural network.

1. The exact SS $s_i$ captures all the information (i.e., required by the mediator) within party $i$'s dataset $\mathcal{D}_i$. Thus, the mediator can do valuation and generate model rewards from the perturbed SS $\{o_i\}_{i \in N}$ without requesting more information from the parties. This limits the privacy cost and allows us to rely on the DP post-processing property.

2. In Sec. 3, the proof that Def. 3.1 satisfies a privacy-valuation trade-off (V3) uses the properties of SS.

Our work introduces privacy as an incentive and simultaneously offers a new perspective that excessive DP can and should be deterred by introducing privacy-valuation and privacy-reward trade-offs and accounting for the DP noise. We use Bayesian models with SS as a case study to show how the incentives and trade-offs can be achieved. It is up to the future work to address the non-trivial challenge of ensuring privacy-valuation and privacy-reward trade-offs for other models.

### Question 4: Can alternative fair reward schemes be used in place of $\rho$-Shapley fair reward scheme [47]?

**Answer:** Yes, if they satisfy P3 and P4. For example, if the exchange rate between the perturbed SS quality and monetary payment is known, then the scheme of [40] can be used to decide the reward instead. Our work will still ensure the privacy-valuation trade-off and provide the mechanism to generate the model reward $q_i(\theta)$ to attain any target reward value $r_i^*$ while preserving similarity to the grand coalition $N$'s model (P5).

### Question 5: What is the difference between our work here and that of [47]?

**Answer:** We clearly outlined our contributions in bullet points at the end of the introduction section (Sec. 1) and in Fig. 1.

At first glance, our work seems to only add a new privacy incentive. However, as discussed in the introduction section (Sec. 1), privacy is barely considered by existing collaborative ML works and raises significant challenges. The open questions/challenges in [64]'s survey on adopting DP in game-theoretic mechanism design (see Sec. 7.1 therein) inspire us to ask the following questions:

- How can DP and the aims of cooperative game theory-inspired collaborative ML be compatible? Will DP invalidate existing properties like fairness?
- How should parties requiring a strong DP guarantee be prevented from *unfairly* and *randomly* obtaining a high-quality model reward?

We propose to enforce a *provable* privacy-valuation trade-off to answer the latter. The enforcement involves novelly selecting and combining the right valuation function and tools, such as DP noise-aware inference.

Additionally, we propose a new reward control mechanism that involves tempering the likelihood (practically, scaling the SS) to preserve similarity to the grand coalition's model (P5) and hence increase the utility of the model reward.

### Question 6: Will a party with high-quality data (e.g., a large data quantity, less need for DP guarantee) be incentivized to participate in the collaboration?

**Answer:** From Fig. 3, it may seem that a rich party $i$ with ample data and a weak privacy guarantee (i.e., large $\epsilon_i$) has a lower utility of model reward to gain from the collaboration. However, it may still be keen on a further marginal improvement in the utility of its model reward (e.g., increasing the classification accuracy from $97\%$ to $99\%$ and predicting better for some sub-groups) and can reasonably expect a better improvement as other parties are incentivized by our scheme (through enforcing a privacy-valuation trade-off and fairness F4) to contribute more data at a weaker yet satisfactory DP guarantee (see App. H.5). Moreover, a rich party does not need to be concerned about others unfairly benefiting from its contribution as our scheme guarantees fairness through Shapley value. In Fig. 8, as a party selects a weaker DP guarantee (and all

else being held constant), the Shapley values of others, which determine their model rewards, decrease.

## Question 7: What is the impact of varying other hyperparameters?

**Answer:** The work of [47] proposes $\rho$-Shapley fairness and theoretically and empirically show that any $\rho > 0$ guarantees fairness across parties and a smaller $\rho$ will lead to a higher attained reward value $r_i$ for all other parties which do not have the largest Shapley value. These properties apply to our problem setup, and using a larger $\rho$ will worsen/reduce $r_i$ and the utility of party $i$'s model reward $q_i(\theta)$ measured by MNLP$_r$. The work of [47] has empirically shown that the number of parties does not impact the scheme's effectiveness. However, it affects the time complexity to compute the exact and approximate SV.

More importantly, the extent to which party $i$ can benefit from its contribution depends on the quantity/quality of its data relative to that of the grand coalition $N$ (and the suitability of the model or informativeness of the prior).

Party $i$'s DP guarantee $\epsilon_i$ is varied in Sec. 6 while the DP guarantee $\epsilon_k$ of the other party $k$ and its number $c_k$ of data points for $k \in N$ are varied in App. H.5. The privacy order $\lambda$ is varied in App. H.5. Across all experiments, we observe that the privacy-valuation trade-off holds. Moreover, when (i) a party $i$ has lower-quality data in the form of fewer data points or smaller $\epsilon_i$, or (ii) another party $k$ has higher-quality data such as a larger $\epsilon_k$, the improvement in the utility of its model reward over that of its individually trained model is larger.

## Question 8: Can privacy be guaranteed by using secure multi-party computation and homomorphic encryption in model training/data valuation?

**Answer:** These techniques are designed to prevent direct information leakage and prevent the computer from learning anything about the data. However, as the output of the computation is correct, any mediator and collaborating party with access to the final model can query the model for predictions and infer private information/membership of a datum (indirect privacy leakage). In our work here, every party can access a model reward. Hence, the setup should prevent each party from inferring information about a particular instance in the data beyond what can be learned from the underlying data distribution through strong *DP guarantees*.

## Question 9: In Sec. 4, we mention that (i) it is possible to have negative marginal contributions (i.e., $v_{C \cup i} < v_C$) in *rare* cases and (ii) adding some noise realizations may counter-intuitively create a more valuable model reward (e.g., $r_i > v_N$). Why and what are the implications?

**Answer:** For our choice of valuation function via Bayesian surprise, the party monotonicity (V2) and privacy-valuation trade-off (V3) properties involve taking expectations, i.e., *on average/in most cases*, adding a party will not decrease the valuation (i.e., the marginal contribution is non-negative), and strengthening DP by adding more noise should decrease the reward value. However, in rare cases, (i) and (ii) can occur. We have never observed (i) in our experiments, but a related example of (ii) is given in Fig. 14a: A larger $\tau_i$ surprisingly increased the valuation.

The implication of (i) is that the Shapley value $\phi_i$ may be negative, which results in an unusable negative/undefined $r_i^*$. However, this issue can be averted while preserving P3 by upweighting non-negative MCs, such as to the empty set, as mentioned in Footnote 10. The implication of (ii) is that some (large) noise realization can result in a more valuable model reward than the grand coalition's model, i.e., $r_i > v_N$. However, collaborating parties still prefer $p(\theta|\boldsymbol{o}_N)$ valued at $v_N$ as the more surprising model reward is *not* due to observations and information. This motivates us to define more specific desiderata (P1 and P2) for our reward scheme.

Lastly, one may question if we should change the valuation function. Should we use the information gain $\mathbb{I}(\theta; \boldsymbol{o}_C) = \mathbb{E}_{\boldsymbol{o}_C}[v_C]$ on model parameters $\theta$ given perturbed SS $\boldsymbol{o}_C$ instead to eliminate (i) and (ii)? No, the information gain is undesirable as it disregards the *observed* perturbed SS $\boldsymbol{o}_C$ and will not capture a party's preference for higher similarity of its model reward to the grand coalition $N$'s posterior $p(\theta|\boldsymbol{o}_N)$.

