# OpenReview forum: "Incentives in Private Collaborative Machine Learning"
_NeurIPS.cc/2023/Conference — NeurIPS 2023 poster_

### Official Review · Reviewer_oNeh · 2023-06-29

**Soundness:** 3 good
**Presentation:** 3 good
**Contribution:** 3 good
**Rating:** 7
**Confidence:** 3

**Summary:**

This paper studies the problem of learning a machine learning model collaboratively under differential privacy and the incentives for parties to participate in the effort. More specifically, in the papers setting, multiple parties are sharing private sufficient statistics to a central aggregator. DP induces noise to the sufficient statistics and parties with less strict privacy guarantees contribute more signal to the inference than the ones with strong privacy guarantees. Therefore, authors argue that there should be an incentive structure in participating in the scheme which rewards the parties with less strict privacy guarantees more. Besides accuracy, authors consider fairness as another desired property from the collaboration. Authors propose a method, that learns a posterior distribution for the parameter of interest using the shared private suff. stats., and releases tempered posterior samples to each client. The level of tempering will determine how close the samples are to the true posterior, and parties with stricter privacy guarantees get samples from more flattened (less tempered?) posterior. The level of tempering is selected by the central aggregator based on the Bayesian surprise which measure how much more information the parties data brings in (measure in terms of the KL divergence against the prior). Authors demonstrate empirically, that a party's reward from collaboration is a better model than simply using their own posterior learned from local data under DP. Furthermore the results show that the reward mechanism is able to give larger rewards for looser privacy guarantees. Finally, authors demonstrate that the proposed reward control mechanism outperforms an existing one (i.e. gets larger benefits from the collaboration), especially when the reward is low.

**Strengths:**

I believe the setting where multiple data holders would like to collaboratively learn from sensitive data is very realistic, and proposing a reasonable incentive structure for participation is a valuable goal. The proposed structure, that promises more reward for less perturbed data, sounds reasonable to me. The theoretical properties (especially V2 and V3) of KL as a valuation function are novel as far as I know, and give a solid theoretical foundation for using it. Using tempering to control the reward is also novel and sounds like a reasonable choice.

The empirical evaluation over multiple models and data sets gives support for the incentive structure, showing that a party can gain from the collaboration and as the $\epsilon$ grows they are rewarded more.

**Weaknesses:**

The method relies on a rather strong trust model that the parties contribute honestly to the collaboration. As the surprise, which determines the valuation for a party, is computed as the KL-divergence between the parties posterior and the prior, it would be quite easy for a party to send bogus data to the mediator that would yield a large KLD (especially if the prior is know). Authors do address this limitation in Remark in line 172 (and extensively in Appendix I), but I would encourage authors to extend this discussion in the main paper as well.

**Questions:**

- You say that you can optimize the $\kappa_i$ to control $q_i(\theta)$ using any root-finding algorithm. However, to solve the $q_i(\theta)$ for a given $r_i^*$ you would need to optimize the KL-divergence between the tempered posterior and the prior right? As you say in Section 3, this KLD is intractable, and hence you need to deploy MCMC methods to evaluate it. How computationally expensive is this? Would you need to run the entire MCMC chain for each $\kappa_i$ value from scratch?
- Figures 2a and 2c: is the difference between $v_2$ and $v_N$ on the largest $\epsilon$ due to some numerical issue in valuating the party? Or are they supposed to overlap if $r_2 = v_N$?
- Minor comment: On line 238, you say that "... party k will now enjoy $(1/\tau)$-DP guarantee ...". I guess you don't really mean this as pure DP guarantee, but as an RDP guarantee? If so, it would be good to add $\lambda$ in the privacy notation (and call it RDP instead of DP).

**Limitations:**

The limitations of the method are discussed in the main paper and more thoroughly in the appendix.

---

> ### Author Rebuttal · Authors · 2023-08-07
>
> Thank you for your detailed summary of our paper and comments. We will address some of the weaknesses and questions below and include them in the revision of our paper.
>
> > Weaknesses.
>
> Thank you for referring to Appendix I for our discussion on the truthfulness assumption. We will find the space to transfer part of the discussion to the main paper.
>
> > Q1 on optimising $\kappa_i$
>
> Yes, we run the entire MC sampling for each $\kappa_i$ value from scratch. The computational complexity is given in Appendix F. Importantly, the number of runs is only a constant factor of Step 3.
>
> We think that our mechanism is computationally practical and can be scaled to more parties.
> When there are more parties, the mediator can sample coalitions according to [25, M, Z] to approximate the Shapley value (in step 3 of App F) within a desired absolute error.
>
> In the reward phase (step 5 of App F), the  $(\kappa_i, r_i)$  pairs evaluated for root-finding of party $i$ can be reused to identify a narrower range for root finding for the other parties, thus reducing the number of evaluations.
>
> [M] Maleki, S., Tran-Thanh, L., Hines, G., Rahwan, T., & Rogers, A. (2013). Bounding the estimation error of sampling-based Shapley value approximation. arXiv:1306.4265, 2013.
>
> [Z] Zijian Zhou, Xinyi Xu, Rachael Hwee Ling Sim, Chuan Sheng Foo, and Bryan Kian Hsiang Low.Probably Approximate Shapley Fairness with Applications in Machine Learning. AAAI, 2023.
>
> > Figures 2a and 2c:
>
> In Fig. 2a and 2c, for the largest $\epsilon$, we have (1) $r_2 = v_N$ and (2) party 2 model reward's value $r_2$ is slightly larger than the value of its perturbed SS $v_2$.
> Recall that the largest $\epsilon$ corresponds to weaker privacy and thus more information in $o_2$.
>
> (1) As party 2 has the highest Shapley value, by fairness P3, party 2 should get a higher reward value than others. By P2, party 2 should get a model reward with value $r_2 = v_N$.
>
> (2) Party 2 model reward's value $r_2$ is larger than the value of its perturbed S $v_2$ as the model reward is additionally trained on the data from party 1 and 3. However, as party 1 and 3 selected moderate privacy guarantee $\epsilon_1 = \epsilon_3 = .2$, the increase is small.
>
> > Minor comment
> Yes, it should be the RDP guarantee. We will make the correction in the paper!

---

> > ### Comment · Reviewer_oNeh · 2023-08-16
> >
> > Thanks for addressing my questions! I'm happy to keep my score as is.

---

### Official Review · Reviewer_Rv9L · 2023-07-07

**Soundness:** 4 excellent
**Presentation:** 4 excellent
**Contribution:** 3 good
**Rating:** 7
**Confidence:** 1

**Summary:**


This research paper investigates the intersection of data sharing incentives and privacy concerns within the realm of collaborative machine learning (ML). Collaborative ML aims to improve model quality by leveraging diversified data from multiple parties, yet the potential benefits of this practice are often hampered by concerns about privacy and the costs of sharing data.
Several existing studies have acknowledged the need for incentives that encourage collaboration, such as guaranteed fair rewards for valuable data contributions. Yet, these incentive-based rewards expose the parties to privacy risks. Meanwhile, some solutions enforce differential privacy (DP) to mitigate privacy concerns but may inadvertently compromise the perceived fairness of collaboration and group welfare. This research fills the gap by proposing an incentive-aware, privacy-preserving reward scheme.
The authors address several questions in their investigation. They consider the impact on valuation and reward if a party demands stronger DP guarantee, suggesting that such a party's reward should generally decrease to avoid randomness due to DP noise. They propose a privacy-valuation trade-off and explore ways to value a party's data, focusing on the quality of inference of model parameters under DP. Finally, they detail a reward scheme designed to maintain privacy, individual rationality, and fairness.

The paper's significant contributions include the development of a new privacy-valuation trade-off criterion, the proposal of novel incentives, and the introduction of reward control mechanisms to adjust the distribution of posterior samples of model parameters among different parties. These solutions aim to preserve similarity to the grand coalition's model while deterring excessive DP.

**Strengths:**

A key strength of this paper is its addressing of a highly significant, real-world issue. Collaboration between diverse parties is a critical aspect of machine learning. However, progress in this field is often hindered by concerns surrounding privacy. Therefore, the paper's proposition of a framework that integrates privacy within the incentive scheme of a collaborative process is not just beneficial, but essential. It effectively merges theoretical constructs with pragmatic applications, fostering advancements in secure, collaborative machine learning practices. Moreover, the paper has good organization and clarity.


**Weaknesses:**

While the paper might appear to lack novelty as it adopts numerous techniques from previous work in collaborative machine learning, it does so within the context of differential privacy. This presents a unique perspective. The process of designing incentives that take into account privacy considerations necessitates an innovative approach.

**Questions:**

no questions

**Limitations:**

yes

---

> ### Author Rebuttal · Authors · 2023-08-07
>
> Thank you for your encouraging feedback! We appreciate your detailed and accurate summary of the contribution of our work and the strengths. We also appreciate that the reviewer has recognized the novelty of our work as the process of designing incentives that take into account privacy considerations. This necessitates an innovative approach combining numerous techniques, which is the main thrust of this paper.

---

### Official Review · Reviewer_kKLU · 2023-07-10

**Soundness:** 3 good
**Presentation:** 3 good
**Contribution:** 2 fair
**Rating:** 6
**Confidence:** 4

**Summary:**

This paper proposes a mechanism to do single-round private collaborative model learning by several agents, each with access to a dataset. The agents do not want to share their data and instead exchange perturbed sufficient statistics of their data, which the central server must aggregate and learn from. Since the devices benefit from collaboration, they might be incentivized to add excessive noise to their sufficient statistics to get the maximum privacy possible while still getting some benefit from collaboration. While this might benefit specific agents, it might not be the ideal scenario from the perspective of the majority of agents. This paper's mechanism disincentivizes this behavior by providing different models to each client, based on their differential privacy requirements, with better models to agents with "most useful" sufficient statistics. The rewards provided by the server also satisfy other desirable properties such as fairness, individual rationality, group welfare, etc.

Overall I like the paper. It is a good step towards reconciling federated learning with agents' strategic behavior. Apart from complaints about presenting the overall scheme, and other minor writing issues, I do not have major concerns about the paper and recommend accepting it. I would recommend the authors incorporate the suggestions below to improve the exposition. I am open to increasing my score.


**Strengths:**

The paper addresses a significant problem: agents' strategic and self-serving behavior in federated learning. Federated learning hopes to enable large-scale privacy-preserving collaborations and encourage users to share the benefits of their data to develop an overall better model. Most incentive designs work with explicit monetary rewards, but without an actual server, that is not doable. This paper bypasses the issue and designs model rewards that satisfy several desirable properties. The paper is also quite exhaustive in its treatment of which desirable properties must be satisfied at each step.

**Weaknesses:**

1. The paper is written in a bottom-up approach, i.e., from sufficient statistics to the final model. It is OK to write it this way, but the overall scheme of things is unclear after reading the paper. I know figure one is an attempt to summarize everything, but an actual pseudocode would be helpful to show explicitly which quantities are being computed, how sampling occurs, etc. It would be helpful to also indicate in the pseudocode which sampling steps are approximated using MCMC procedures. For instance, currently, the way it's presented is unclear up to section four; why is the paper talking about coalitions? An informed reader might guess it is due to Shapley value computations, but otherwise, it seems a bit arbitrary. Writing the paper top-down or adding the above pseudocode might clarify these aspects.

2. The individual rationality definition in P4 seems incorrect. Why would the agent perturb the sufficient statistic if they are alone? Collaboration with perturbed statistics should be better than unperturbed individual effort.

3. Finally, the complexity of computation/sampling will also become explicit by being more precise about the exact computations. Shapley value computations are generally expensive, and discussing the effect of inexactness in the procedure would be good.

**Questions:**

See comments above. Can actual individual rationality be satisfied by the mechanism?

**Limitations:**

The paper should emphasize the scope of the work. Most federated learning happens over several interactions and involves local processing on the agent. There might be some applications where a single round of interaction between the server and the clients is enough, but that is an exception, not the norm. It is unclear if the mechanism provided in this paper can be applied repeatedly in some manner. I understand this extension might be complex, but not discussing it is incorrect. I urge the authors to list potential applications they have in mind in the motivation. As mentioned above, the authors should be more explicit about the exact computations and the complexity of running the mechanism. Finally, the individual rationality requirement in P4 seems wrong. If this can not be changed, it should be emphasized strongly that this is a limitation of the approach.

---

> ### Author Rebuttal · Authors · 2023-08-07
>
> We thank you for your helpful comments and suggestions! We have responded to them below and hope it will improve your opinion of our work.
>
> >Weakness 1
>
> Our paper is written in a bottom-up approach and defers information to when it is needed or the appendix: App.A.3 contains the pseudocode for sampling, App.B (lines 684-94) contains information for readers less familiar with data valuation and collaborative ML, and App.F describes the main steps of our scheme and how various quantities should be computed. We agree that it is useful to give an overview (e.g., to clarify why coalitions are mentioned) in the main paper and will add one to Sec.2.
>
> We have attached an overview of the quantities and steps involved in our private collaborative ML scheme in the global pdf.
>
> >Weakness 2
>
> An agent will perturb the sufficient statistic (SS) when alone to protect the privacy of data owners from curious users of its ML model. For example, a hospital would not want its doctors to infer much about any patient’s data and a firm would not want employee users to infer about customers. This motivation applies to existing DP works like DP-SGD [Abadi, 2016] and DP noise-aware inference [Bernstein & Sheldon, 2018; Kulkarni et. al, 2021]. We will clarify the above in our paper.
>
> We think that P4 is correct as (1) from the mediator's perspective, party $i$'s submitted perturbed SS is only worth $v_i$ and rationality is defined according to conventions of game theory, (2) each party may still want DP when alone, and (3) it seems natural for $i$'s reward value $r_i$ to be less than the value of its exact SS $s_i$ if $i$ selected an excessively strong DP guarantee (e.g., and collaborator $j$ only has one data point).
>
> > Can actual individual rationality (AIR) be satisfied by the mechanism?
>
> We assume that AIR means that each party $i$'s reward value $r_i$ is at least the value of its unperturbed SS $s_i$. AIR may be satisfied when parties are incentivized enough to select a large ϵ (weak DP). However, AIR cannot be theoretically guaranteed by the mechanism as parties are still free to seek stronger DP (footnote 2) that reduces the benefit of the collaboration and the mediator cannot access the private $s_i$ to generate $i$'s model reward.
> The mediator should use party $i$'s perturbed SS $o_i$ to generate its model reward to incentivize $i$ to be truthful and contribute more valuable information (Q2 in App.I).
>
> However, P4 can be theoretically guaranteed if the $𝜌$ used in the $𝜌$-Shapley fairness scheme is $\leq 𝜌_r$ defined in Sim et al's Theorem 1. We will highlight that the stronger AIR has not been theoretically guaranteed as a limitation.
>
> >Weakness 3
>
> We have been more precise about the exact computations and time complexity in App.F. In our revision, we will further reference the citations/pseudocode for computing the local SS, clarify that step 3 has to be repeated for each coalition $C \subseteq N$, and step 5 uses the scaled perturbed SS $\kappa_i o_j, \kappa_i c_j, \kappa_i Z_j$ for each party $j \in N$ for noise-aware inference.
>
> We would add that: when there are <6 parties (as in our experiments), it is feasible to compute the Shapley value (SV) exactly. When there are more parties, SV has to be approximated and the inexactness (absolute error in the SV estimate) can be controlled by sampling enough coalitions and in accordance to steps outlined by [25, M, Z].
>
> [M] Bounding the estimation error of sampling-based Shapley value approximation. arXiv:1306.4265, 2013.
>
> [Z] Probably Approximate Shapley Fairness with Applications in Machine Learning. AAAI 2023.
>
>
> > Limitations: The paper should emphasize the scope of the work.
>
> In Sec. 2 & 8, we clearly stated that our work only covers the scope of Bayesian models with SS.
>
> Our method would still work if a party submitted perturbed SS in _multiple_ rounds of interaction instead. The value of party $i$ (coalition $C$) should be the Bayesian surprise of party $i$'s (coalition $C$'s) SS from all rounds and party $i$'s model reward involves scaling all perturbed SS across rounds by $\kappa_i$. The mediator can reward each party only once at the end of the collaboration and make either of the following modifications to use our scheme:
> - Sum the perturbed SS across rounds for each party $i$ (due to Line 101)
> - Replace $o_N$ on Line 261 with the set of perturbed SS across rounds and modify noise-aware inference (Algo 1) by adding an extra for loop for different rounds
>
> We will include this discussion in our revised paper.

---

> > ### Comment · Reviewer_kKLU · 2023-08-14
> >
> > > We have attached an overview of the quantities and steps involved in our private collaborative ML scheme in the global pdf.
> >
> > Thanks, this is indeed helpful.
> >
> > > An agent will perturb the sufficient statistic (SS) when alone to protect the privacy of data owners from curious users of its ML model. For example, a hospital would not want its doctors to infer much about any patient’s data and a firm would not want employee users to infer about customers.
> >
> > Please clearly discuss scenarios where the data owner will also perturb the statistics individually in the revised version. This is an important point.
> >
> > Can the paper's guarantees be extended to the setting with the stronger individual rationality definition, i.e., when a single agent does not perturb the statistics? From the response, it seems not. Note that while studying privacy-utility trade-offs in federated learning, a collaborative algorithm is usually expected to perform better than the non-collaborative baseline, which does not have to care about privacy. For instance, while doing collaborative Gaussian mean estimation with non-strategic agents, assuming low data heterogeneity, agents will benefit from collaboration as long as the privacy noise level is comparable to the inherent variance of a consensus mean estimate. This was the intuition behind my question: can the work highlight the utility of collaboration when agents do not need to add any noise on their own?
> >
> > > We have been more precise about the exact computations and time complexity in App.F. In our revision, we will further reference the citations/pseudocode for computing the local SS, clarify that step 3 has to be repeated for each coalition ....
> >
> > This would indeed improve the presentation, thanks.
> >
> > > The mediator can reward each party only once at the end of the collaboration and make either of the following modifications to use our scheme:
> >
> > I am not sure I understand the scheme and if it clarifies my concern. When I said that interaction happens across several rounds in most federated learning applications, I meant the agents also get to see a sequence of models instead of just one final model. From what I understand, the mediator only gives the agents a single model reward even with the modification. This is a collaborative learning model, but I am unsure if calling this setup federated learning is appropriate. This seems more akin to distributed estimation literature with privacy constraints. The real challenge with maintaining privacy across multi-round interactions is adding more noise which might lead to a worse privacy-utility trade-off. Unfortunately, most works dealing with incentives in federated learning do not consider a multi-round interaction.

---

> > > ### Author Response · Authors · 2023-08-15
> > >
> > > Thank you for your quick and detailed reply! We will respond to your follow-up questions below.
> > >
> > > > Please clearly discuss scenarios where the data owner will also perturb the statistics individually in the revised version. This is an important point.
> > >
> > > Thanks for the suggestion. We will better clarify why and when the data owner will perturb the statistics individually in the revised version.
> > >
> > > > Can the paper's guarantees be extended to the setting with the stronger individual rationality definition, i.e., when a single agent does not perturb the statistics? can the work highlight the utility of collaboration when agents do not need to add any noise on their own?
> > >
> > > We believe you are clarifying if we can guarantee stronger individual rationality (SIR/AIR) , i.e., the model reward $q_i(θ)$ trained on perturbed SS is more valuable than the party $i$’s model trained on its exact SS $s_i$. The short answer is no. To add to our previous response on AIR, the mediator only incentivizes the participants against selecting excessively strong DP guarantees. As the mediator does not restrict the maximum DP noise added by each party, the mediator cannot control the privacy-utility tradeoff and guarantee SIR. Guaranteeing SIR is a non-trivial challenge left for future work.
> > >
> > > However, we have considered the following modification to the reward mechanism to guarantee SIR. Instead of rewarding model parameter samples, the mediator can reward each party with perturbed SS $t_j^i$ (for Sec. 5.1)  or $κ_i \boldsymbol{o}_j, κ_i c_j, κ_i Z_j$ (for Sec. 5.2) for every other party $j$. Then, each party $i$ is free to use its rewards and its own unperturbed SS $s_i$ for inference, thus achieving SIR. However, we did not go with this alternative mechanism as it faces incentive issues — as party $i$’s model reward would not be directly influenced by its submitted $o_i$, it may be less deterred (hence more inclined) to submit less informative or fake SS (see Question 2 in App. I).
> > >
> > > Thanks for identifying an interesting and important point about strong IR. We will empirically evaluate if strong IR has been achieved, and include the theoretical limitation and above discussion in our paper. We wish to convey that the limitation is acceptable when parties care about privacy even when alone. Even when parties do not, the limitation is needed to incentivize parties to submit (i) informative and real perturbed SS that they are willing to use, while (ii) not compromising for weak DP guarantees.
> > >
> > > > When I said that interaction happens across several rounds in most federated learning applications, I meant the agents also get to see a sequence of models instead of just one final model. From what I understand, the mediator only gives the agents a single model reward even with the modification.
> > >
> > > Your understanding of our rebuttal is correct --- we suggested that the mediator gives the agents a *single* model reward after aggregating SS across rounds.
> > >
> > > To clarify the second modification in our rebuttal on how to use our mechanism repeatedly,  consider two parties (subscripted) who take part in $t$ rounds (superscripted). At each round $t$, each party will only submit perturbed SS generated from _new_ data. The mediator can  use Algo1 to compute the noise aware posterior $p(θ|o^1_1, \ldots, o^t_1, o^1_2, \ldots, o^t_2)$ and use it to replace $p(θ|o_1, o_2)$ from the single-round setting. Valuation and reward can be done as before. Note that although more noise is added in the single round setting to generate $o^1_i, \ldots, o^t_i$, the mediator has more fine-grained information from considering them as separate variables.
> > >
> > > Interestingly,  the post-processing property of DP will allow the mediator to use the perturbed SS $o^i_t$ from round $i$ to generate model rewards in rounds $i, \ldots, t$ without any additional privacy leakage.
> > >
> > > > This is a collaborative learning model, but I am unsure if calling this setup federated learning is appropriate.
> > >
> > > We agree with you that our focus on Bayesian models with SS does not fit the standard federated learning setup. In the paper, we have reiterated the focus on models with SS and “collaborative ML” in Sec. 2 and 8. Although we discussed multiple rounds above, we will state our focus on the single-round setting in our revision.
> > >
> > > > Challenge with maintaining privacy across multi-round interactions
> > >
> > > We agree that in a multi-round interaction, agents may have to add more noise to their sufficient statistics and it may lead to a worse privacy-utility trade-off than a single round setting. We have also identified other challenges dealing with incentives in the multi-round setting in Sec. 7 on related works (lines 369-376). This motivated us to focus on models with sufficient statistics as SS can be easily aggregated (lines 91-93) instead.
> > >
> > > Once again, thank you for the suggested revisions! We hope that we have clarified your concerns and we are happy to answer further questions.

---

> > > > ### Comment · Reviewer_kKLU · 2023-08-16
> > > >
> > > > Thanks for the clarifications. I hope you will sincerely discuss the abovementioned limitations and contextualize your work well in the existing literature. I will increase my score by a point conditioned on these modifications.

---

> > > > > ### Author Response · Authors · 2023-08-17
> > > > >
> > > > > Thank you for acknowledging our responses and the constructive suggestions! We will update our revision to further contextualise our work and include the other discussion (e.g., stronger rationality) in the appendix.

---

> > > > > > ### Comment · Reviewer_kKLU · 2023-08-18
> > > > > >
> > > > > > I would encourage at least describing the choice of P4 in the main paper, not the appendix, along with a simple example. This can be done with a remark in a few lines.

---

> > > > > > > ### Author Response · Authors · 2023-08-19
> > > > > > >
> > > > > > > Thanks for the suggestion! We will do that in the revision.

---

### Official Review · Reviewer_zMuU · 2023-07-23

**Soundness:** 2 fair
**Presentation:** 4 excellent
**Contribution:** 2 fair
**Rating:** 6
**Confidence:** 3

**Summary:**

From a mostly empirical angle, the paper studies a new valuation metric for incentivizing agents to share their data for collaborative ML while ensuring the data they share is Renyi-DP. Particularly, the KL-divergence between agents' prior and posterior, or the Bayesian surprise, is used as the value of the (possibly perturbed) observations received by the agents. Finally, the proposed method is validated on a synthetic dataset, a regression dataset, and a classification dataset.

**Strengths:**

1. The experimental results are explained carefully, in detail, and the figures emphasize the key takeaways from the paper.
2. Overall the paper is easy to follow and nicely presented.
3. Ethical concerns and potential societal impact are discussed in depth, and in detail, in the appendix.
4. The problem being studied is an interesting one and especially timely given current conversation on privacy and ethical concerns of large scale ML models.


**Weaknesses:**

1.  While the paper is mostly empirical and should be assessed accordingly, the proposed method still lacks mathematical justification. For instance, it is unclear for what choices of $\rho$ (or if any) would IR be satisfied for all agents. Quantitative analysis of the desiderata for different values of $\rho, \lambda, \epsilon$, especially quantifying the interplay between $\rho$ and $(\lambda, \epsilon)$ Renyi-DP, would strengthen the results. At the current stage, it is unclear under what kind of problem settings or privacy constraints would the proposed mechanism be feasible.
2.  In the same vein, desiderata **P5** and **P6** are only discussed intuitively. For ML practitioners with money at risk, additional discussions (and perhaps mathematical or game-theoretic justifications) would be more persuasive.
3.  While Appendix I, Q5 has partially addressed the relationship between the work and [Sim et al., 2020], as discussed above, it is not clear how **P5** fit in the existing theoretical framework around Shapley values. Similarly, while the tempering technique has not been introduced in prior works, the mathematical or game-theoretic implications of $\kappa$ are not addressed, and only an intuitive discussion is provided in the appendix.
4.  The potential application of the method seem limited in domain. The method does not appear to be generalizable outside of Bayesian linear regression. Moreover, calculating the proposed values already entails a costly MCMC inference procedure in order to evaluate the Baeysian surprise. It is unclear if such procedures would remain practical even for smaller neural nets. (While the appendix has discussed various approximation schemes, it may be difficult to persuade practitioners to adopt the method without additional mathematical guarantees on how robust the proposed desiderata are to the estimation errors in both KL and posterior).
5.  Desiderata **P2** is a strict relaxation of the weak efficiency constraint (R3) in [Sim et al., 2020]. Particularly it requires only one agent, as opposed to one agent in each group, to take full advantage of shared data, and may discourage agents from participating in the mechanism (especially with limited IR guarantees).

- [Sim et al., 2020]. "Collaborative machine learning with incentive-aware model rewards." International conference on machine learning. PMLR, 2020


**Questions:**

1.  Can the authors explain **P1** in more detail? From my understanding, it seems like **P1** means that agents&rsquo; reward cannot reveal too much information about other agents&rsquo; data. It would be great if the authors could confirm this. It might be me but currently the definition looks a bit confusing.



**Limitations:**

The authors have discussed in detail the potential negative societal impact.

---

> ### Author Rebuttal · Authors · 2023-08-07
>
> Thanks for your detailed review & questions! We will address your concerns below and include them in the revised paper. We hope our clarifications will improve your opinion of our work.
>
> > W1: Math justifications
>
> To clarify, our work includes both mathematical and empirical justifications of the proposed method. We have theoretically shown that V1-V3 (Sec.3, App.C) and P1-P2 can be satisfied.
> P3-P4 can be satisfied by using the quantitative analysis of $𝜌$ in Theorem 1 of Sim et al. [2020] since we adopted their $𝜌$-Shapley reward scheme (line 209).
> In our revision, we will
> - refer readers to Sim et al. [2020] for analysis of the impact of varying $𝜌$;
> - clarify that for **any** privacy constraint $(λ,ϵ_i)$ and problem setting (i.e., dataset), IR is satisfied for all agents if $0 < 𝜌 \leq 𝜌_r$ ($𝜌_r$ is defined in Sim et al [2020]'s Theorem 1 and computed based on $v_i$ & $v_N$). Our results should hold for any privacy constraint;
> - elaborate V3: A higher $λ$ or smaller $ϵ_i$ should lead to lower valuation.
>
> There is limited quantitative analysis of the interplay with $(λ,ϵ_i)$ as, like the properties of the dataset, they only affect the reward value and choice of $𝜌$ _indirectly_ by affecting $\\{v_C\\}_{C \subseteq N}$.
> However, empirically, we have shown that IR holds for a range of $ϵ_i$'s (Figs.2,3) and additionally for a larger $λ$ in Fig.11.
>
> > W3: how P5 fit in the existing theoretical framework around Shapley values (SV)
>
> P5 complements the framework. The framework decides the target reward values $r^*_i$; then, collaborative ML works [46,50,ours] propose mechanisms to generate model rewards to realize the target values. Without P5, the ML practitioner may be indifferent to different model rewards $q_i(θ)$ that achieve the same target reward value. As P5 is a secondary criteria that is maximized after other desiderata have been achieved, it does not change existing SV results.
>
> > W3: Mathematical implication of $κ_i$
>
> In App.E.2, we mathematically prove that a smaller $κ_i$ decreases the Bayesian surprise.
> In our revision, we will cite existing works (on likelihood tempering, power posterior, e.g., https://andrewcharlesjones.github.io/journal/power-posteriors.html) that discuss the implication/interpretation of varying $κ_i$ (e.g., synthetically reducing the dataset size).
> In App.'s Fig.7, we empirically show that likelihood tempering leads to better similarity to the grand coalition’s $p(θ|o_N)$.
>
> > W2: P5 & P6 justifications
>
> **P5** We introduce $r'$ to address parties' secondary preference of similarity to the posterior $p(θ|o_N)$ and a model reward that makes similar predictions. $r'$'s definition is inspired by lines 158-60 & 724 and should decrease with stronger privacy/less data.
>
> A preference for lower KL divergence between the posterior and $q_i(θ)$
> - has precedent (e.g., it is minimised by expectation propagation in Bayesian ML);
> - is empirically justified as we observe that for the same $r_i$, higher similarity (purple line) with $p(θ|o_N)$ in Figs.2f,12b,13b leads to a lower MNLP$_r$ (higher model utility) in Figs.3d-f.
>
> **P6** Maximizing group welfare is a common concept in game theory & collaborative ML [46,50,D]. From party $i$'s perspective, a higher group welfare will either (i) increase $i$'s reward value or (ii) others' reward value. (i) is desired and (ii) is acceptable to party $i$ when fairness is still ensured.
>
> If the above is insufficient, can you clarify what are the intended justifications?
>
> [D] Optimality and stability in federated learning: A game-theoretic approach. NeurIPS 2021.
>
> > W4: Applications
>
> We described in lines 91-3 that our work applies to Bayesian models with SS, not just linear regression. We considered logistic regression in our experiments.
>
> We acknowledge that our method cannot be directly applied to neural networks. However, it may be applicable when ML practitioners manage to generate SS. For example, ML practitioners tend to use existing large pre-trained models (e.g., VGG-16 for images) and only fine-tune the last layer(s), so the linear/logistic model would still be useful.
>
> > Additional mathematical guarantees on how robust the proposed desiderata are to the estimation errors in both KL and posterior
>
> P1&2 will always hold. P3 and P4 may be affected by the errors in KL and posterior estimation.
>
> We will justify why the errors can be low:
> - In App.C.3, we extensively discuss how the error in KL estimation can be reduced by taking more samples.
> - Though the DP noise-aware inference works [Bernstein and Sheldon, 2019; Kulkarni et al, 2021] did not provide theoretical results about the estimation errors, they have empirically demonstrated the similarity to the non-private posterior.
> - Empirically, we checked MCMC diagnostics (e.g., R-hat) to ensure that convergence to the true posterior distribution has been achieved and that the variance of KL estimation across runs is low (Tables 2,3).
>
> > P2 vs. R3 in [Sim et al., 2020]
>
> R3 only allows an agent per group $C$ to take full advantage of the shared data of $C \in CS$ when the coalition structure $CS$ consists of multiple disjoint groups.
> However, when the coalition structure is the grand coalition $CS=\\{N\\}$ (the desired case in Sec.4 Para.2 of [Sim et al, 2020]), only 1 agent can benefit from the shared data of $N$. This is the same as P2.
>
> In our paper, we assume that $N$ will form for simplicity. However, we can consider other coalition structures or ensure the formation by selecting $𝜌 \leq 𝜌_s$ according to Sim et al's Theorem 1.
>
> > Q1. Explain P1.
>
> Your understanding is generally correct.
> To be specific, other agents have already revealed private information with $(λ,ϵ_k)$-DP guarantee to the mediator. Agent $i$’s reward cannot depend on more private information (e.g., ask for more data or samples of model parameters). Instead, it should only use the information already disclosed to the mediator.

---

> > ### Comment · Reviewer_zMuU · 2023-08-16
> > **Update to Rebuttal**
> >
> > Thank you for the detailed feedback! I am mostly convinced by the results. In particular, the discussion on theoretical justifications is greatly appreciated, particularly the part on IR and $\rho$. Looking at only [Sim et al., 2020], it wasn't clear if their results still translate to the setting here and the added discussion will be beneficial.
> >
> > Additional comments on P6: Sorry for the poorly worded and unclear comment. Maximizing sum of rewards (the so-called "social welfare") itself is reasonable and used throughout economic literature. On the other hand, maximizing *total similarity* is not a common objective, nor is it found in prior research on algorithmic game theory. (I also looked into the referenced papers and cannot find *similarity maximization* as a desiderata.) As it cannot be guaranteed that the grand coalition is always formed, it is unclear how would this preference for similarity affect the group welfare.
> >
> > Additional comments on P2, P5, P6: All these assumptions work great when a grand coalition is formed, but this is not always the case. Combined with the fact that P2 only guarantees the performance of a single participant (as opposed to one in each group), the concern is that groups in the "smaller coalitions" will be unfairly disadvantaged. When a grand coalition is not formed, the potential impact of the mechanism's preference for similarity is unclear.
> >
> > I'd gladly raise my score if the you could provide further insights on "the impacts of P[2, 5, 6] when a grand coalition is not formed". Thank you again for the detailed feedback.

---

> > > ### Author Response · Authors · 2023-08-17
> > >
> > > Thank you for responding! We will add the discussion on how Sim et al. [2020]’s results translate to that in our paper in our revision.
> > >
> > > > provide further insights on "the impacts of P[2, 5, 6] when a grand coalition is not formed".
> > >
> > > When the grand coalition does not form, the core idea is that instead of guaranteeing the desiderata based on the grand coalition, we guarantee the original desiderata for each coalition $C$ in the coalition structure $CS$ (*). In addition, for any party $i \in C$, we use the perturbed SS $o_C$ to generate the rewards instead of the grand coalition's $o_N$. This is because $i$ has chosen to only work with $C$ and thus should not use the SS submitted by others.
> > >
> > > We will rewrite P2, P5, P6 for the case when the grand coalition does not form below.
> > > -  **P2.** For any coalition $C \in CS$, there is a party $i \in C$ whose model reward is the coalition $C$'s posterior, i.e., $q_i(\theta) = p({\theta |o_C})$. It follows that $r_i = v_C$ (as in [Sim et al., 2020], R2). One in each coalition (group)  gets the coalition's best model.
> > > - __P5.__  Among multiple model rewards $q_i(\theta)$ whose value $r_i$ equates the target reward $r^*\_i$, we secondarily prefer one with a higher similarity $r'_{i,C} = -D\_{KL}(p(\theta |o\_C);q_i(\theta))$ to the coalition's posterior $p(\theta |o\_C)$ where $i \in C$.
> > > - **P6.** _After_ maximizing the total reward value $\sum^n_{i=1} r_i$, the reward scheme should also maximize the total similarity $\sum^n_{i=1} r'_{i,C}$.
> > >
> > > (*) Note that like in [Sim et al., 2020] , we assume that the coalition structure is known and do not propose how to derive or select the coalition structure. However, we can select $\rho$ according to Theorem 1 to ensure the grand coalition will form.
> > >
> > > > On the other hand, maximizing total similarity is not a common objective, nor is it found in prior research on algorithmic game theory. (I also looked into the referenced papers and cannot find similarity maximization as a desiderata.) As it cannot be guaranteed that the grand coalition is always formed, it is unclear how would this preference for similarity affect the group welfare.
> > >
> > > Sorry for misunderstanding your question previously. Indeed, similarity maximization is non standard and we will explain below how it should not affect the group welfare.
> > >
> > > In the paper, we wrote that _while_ maximizing the total reward value $\sum^n_{i=1} r_i$, the reward scheme should also maximize the total similarity. We will revise the _while_ to _after_.  With this modification, P6 can be satisfied by "maximizing the total reward value $\sum^n_{i=1} r_i$'' and satisfying P5.
> > >
> > > Satisfying P5 would ensure that the mediator selects the reward $q\_i(\theta)$ with higher similarity $r'\_i$ or $r'\_{i,C}$ among rewards with the same reward value $r\_i$.  Thus, it would help to maximize the total similarity $\sum^n_{i=1} r'\_{i,C}$  in P6 for the same group welfare ($\sum^n_{i=1} r\_i$) value.
> > >
> > > As P5/similarity is a secondary criterion  that is maximized after other desiderata have been achieved, it does not change existing SV and group welfare results.  We will consider removing the similarity part in P6 to align with the current literature as it is already implied by P5.
> > >
> > > We have given the modification when the grand coalition does not form above, and the reason for preferring higher similarity in the previous response.
> > >
> > > Once again, thank you for your suggestions that will help to improve the paper. We hope the above clarifications address your concerns and we will be happy to provide further clarifications.

---

> > > > ### Comment · Reviewer_zMuU · 2023-08-17
> > > >
> > > > Thank you for the continued discussion. Please do include the discussion and the restated P[2, 5, 6] in the revision. I have updated my score as promised.

---

### Author Rebuttal · Authors · 2023-08-09

We thank all reviewers for the encouraging feedback that recognizes the novelty of our work. We appreciate the high-quality reviews and valuable feedback which we will consider carefully in revising our paper.  In our rebuttal, we have
- Clarified the (mathematical) justifications of P1-6 (Reviewer zMuU);
- Clarified reviewers’ doubt about the problem setting by
    - Referring the reviewers to Secs. 2 & 8 which stated that our approach works for models with sufficient statistics;
    - Explaining the need for DP and perturbed SS when alone (Reviewer kKLU);
    - Justifying our choice of rationality in P4 (Reviewer kKLU).
- Pointed the reviewers to App.F for a discussion of the computational complexity and main steps of our scheme. We also provided an alternative diagrammatic overview in the PDF attached.

Please let us know if you have any more questions, and we would be happy to address them within our allowed period.

We would like to correct a minor typo in P5: $r’_i$ should be the negated KL divergence instead. Our subsequent theory/experiment results are not impacted.

---

### Decision · Program_Chairs · 2023-09-21

**Decision:**

Accept (poster)

**Comment:**

This paper studies the private collaborative machine learning and its incentive structures.

The reviewers are positive and agree that the problem studied in this paper is very interesting and practical, and the proposed scheme and mechanism are solid and useful. In addition, the reviewers' concerns have been well addressed.

In light of this, I recommend acceptance and encourage the authors to incorporate the suggested changes (of reviewers) into the revision.